# CONFORMALIZED SURVIVAL COUNTERFACTUAL PREDICTION FOR GENERAL RIGHT-CENSORED DATA

**Sijie Ren**[1,2]**, Meng Yan**[4,5]**, Zhen Zhang**[3,5]**, Yinghui Xu**[1]**, Xinwei Sun**[2] *

[1]Artificial Intelligence Innovation and Incubation Institute, Fudan University
[2]School of Data Science, Fudan University
[3]Zhejiang Key Laboratory of Particle Radiotherapy Equipment, Zhejiang Cancer Hospital
[4]Department of Radiation Oncology, Tianjin Medical University Cancer Institute & Hospital
[5]Department of Radiation Oncology (Maastro), Maastricht University Medical Centre+
`sjren23@m.fudan.edu.cn,yanmeng1999@tmu.edu.cn,`
`zhen.zhang@maastro.nl,{xuyinghui,sunxinwei}@fudan.edu.cn,`

## ABSTRACT

This paper aims to develop a lower prediction bound (LPB) for survival time across different treatments in the general right-censored setting. Although previous methods have utilized conformal prediction to construct the LPB, their resulting prediction sets provide only probably approximately correct (PAC)–type miscoverage guarantees rather than exact ones. To address this problem, we propose a new calibration procedure under the potential outcome framework. Under the strong ignorability assumption, we propose a reweighting scheme that can transform the problem into a weighted conformal inference problem, allowing an LPB to be obtained via quantile regression with an exact miscoverage guarantee. Furthermore, our procedure is doubly robust against model misspecification. Empirical evaluations on synthetic and real-world clinical data demonstrate the validity and informativeness of our constructed LPBs, which indicate the potential of our analytical benchmark for comparing and selecting personalized treatments.

## 1 INTRODUCTION

Predicting survival time under specific treatment is of great importance in making correct predictions in high-stakes domains for healthcare Obermeyer et al. (2019); Navarro et al. (2021). For example, predicting the survival time of lung cancer under different radiochemotherapy regimens is crucial for tailoring therapies to individual patients Wilson et al. (2021); Horne et al. (2024), optimizing outcomes while minimizing side effects. This problem can be formulated as predicting the conditional expectations of potential outcomes under different treatment regimes Rubin (2005). However, survival data are often right-censored Klein & Moeschberger (2006); Wilson et al. (2021), meaning the true survival time is not fully observed for some patients. Unlike traditional causal inference, where outcomes are fully observed, right-censored data pose challenges due to the partial information on survival times.

While many previous methods have been proposed to predict the survival function (Cox, 1972; Murphy et al., 1997; Tibshirani, 1997; Gui & Li, 2005; Steyerberg, 2016), they often rely on model assumptions that are difficult to verify, limiting their ability to provide reliable uncertainty quantification. In contrast, providing a Lower Predictive Bound (LPB) for survival analysis, instead of predicting the entire survival function, offers enhanced reliability and robustness. LPBs are particularly effective at handling censoring and provide a conservative estimate, making them more suitable for high-risk decision-making. This is especially crucial in clinical contexts, where overly optimistic predictions can lead to suboptimal or even harmful treatment choices.

To ensure reliable Lower Prediction Bounds (LPBs), conformal prediction methods Vovk et al. (2005) have been applied to right-censored survival data Candès et al. (2023). Recent works have improved the calibration process by incorporating additional data points. For instance, Qi et al. (2024)

---

assigned estimated values to censored data based on their true event times, while Gui et al. (2024) introduced a covariate-dependent, data-adaptive censoring time to account for the heterogeneity in the censoring mechanism for Type-I right-censored data with a PAC-type guarantee. Additionally, Davidov et al. (2025) extended this framework to handle general right-censored data.

However, these works have two main limitations. First, they do not provide an LPB for the treatment effect on survival time. For example, while Candès et al. (2023) predicted counterfactual survival times under different conditions, this was only applicable to cases where the censoring time exceeded a specific threshold. Furthermore, the LPB provided by these works Gui et al. (2024); Davidov et al. (2025) only offers a *Probably-Approximately-Correct* (PAC)-type guarantee, which does not ensure that the prediction is accurate for the entire population. In contrast, the marginal coverage guarantees more reliable and safer predictions across the entire population, including rare and extreme cases, which is crucial in high-stakes clinical scenarios MacDonald et al. (2025).

To bridge this gap, this paper introduces *conformalized survival counterfactuals prediction*, a novel approach providing exact marginally valid LPB for counterfactual survival outcomes for general right-censored data. At its core, this approach provides an upper bound of a pre-specified nominal level $\alpha$, through a non-conformity score defined via the counterfactual quantile regression function. This upper bound is identifiable, and can be further calibrated using weighted conformal prediction (Lei & Candès, 2021; Jin et al., 2023). We can show that such an LPB can achieve the marginal coverage, provided that the weight function can be well estimated. We also provide the doubly robustness property. We demonstrate the validity and the effectiveness of our method on synthetic data and an in-house lung cancer dataset. We observe that the LPB provided by our method is valid and less conservative than other methods. Besides, the LPB under different radiochemotherapy regimens varies across diverse patient populations in lung cancer, offering valuable insights into personalized treatment strategies.

To summarize, the contributions of this paper can be listed as follows:

- *Survival counterfactual prediction:* We propose a new procedure for quantifying the uncertainty of counterfactual survival time predictions under different treatments in general right-censored data. Our procedure establishes an upper bound for the miscoverage rate that can be reliably identified and further calibrated through weighted conformal prediction.

- *Theoretical guarantee:* We provide a distribution-free exact guarantee for the counterfactual prediction set and quantify the error from weight estimation. We also provide the doubly robustness property.

- *Empirical validation:* We validate our procedure using both synthetic and real clinical data. On the synthetic data, we show that our calibration process yields less conservative LPBs while maintaining the desired coverage guarantee. On the clinical data, we demonstrate its effectiveness in identifying optimal treatments across diverse populations.

## 2 RELATED WORK

Conformal prediction Vovk et al. (2005) was widely used for providing a reliable LPB in survival analysis Candès et al. (2023); Qi et al. (2024); Meixide et al. (2024); Qin et al. (2025) and counterfactual inference Lei & Candès (2021); Jin et al. (2023); Candès et al. (2023); Deshpande & Kuleshov (2024). The conformal prediction for survival analysis was first considered by Candès et al. (2023) for the Type-I right-censored data. Qi et al. (2024) assigned a "best-guess" (BG) value as a surrogate of censored data for their true event times, of which performance is heavily affected by the quality of the imputation. Although distribution-free methods have been proposed in Meixide et al. (2024) and Qin et al. (2025) for constructing LPBs in the general right-censored setting, the validity of their bootstrap-based approaches depends on asymptotic results under specific regularity conditions. Gui et al. (2024) offered more informative LPBs for Type-I right-censored data by employing the adaptive cutoff method by tuning a hyperparameter for quantile regression using holdout calibration data to attain the desired coverage rate, and Davidov et al. (2025) extended this framework to the general right-censored setting with a well-designed data selection strategy for calibration, but both with PAC-type guarantees.

However, none of the above works achieves an exact guarantee on general right-censored data. Candès et al. (2023); Gui et al. (2024) addressed survival analysis under a Type-I right-censored

setting, which assumes that the censoring time $C_i$ is known and relies on the availability of $C_i$. Davidov et al. (2025) considered the general right-censored setting, but with a PAC-type guarantee. In this work, we *firstly* establish a conformalized procedure for survival counterfactuals prediction for general right-censored data with exact coverage guarantee of the LPB.

## 3 PRELIMINARY

Throughout the paper, we focus on the potential outcome framework Rubin (1974); Splawa-Neyman et al. (1990) with different treatments. In particular, this paper focuses on the under-treated *general right-censored data* setting. Specifically, suppose we have $\{W_i, X_i, \widetilde{T}_i, e_i\}_{i=1}^N$, where $W, X, \widetilde{T}, e$ denotes the treatment, the vector of covariates, the observed censored survival time $\widetilde{T} := \min(T, C)$, and the binary indicator $e = 1\{T < C\}$, with $T, C$ respectively denoting the true survival time and the censoring time. This means that for each subject, we can observe either the censoring time or the survival time, but not both, depending on which event occurs first. For simplicity, we consider binary treatments, *i.e.*, $W \in \{0, 1\}$.

Denote by $(T(1), T(0))$ the pair of potential outcomes. We assume that

$$(W_i, X_i, T_i(1), T_i(0), e_i) \overset{i.i.d.}{\sim} \mathbb{P}(W, X, T(1), T(0), e).$$

Under the *stable unit treatment value assumption* (SUTVA) condition Rubin (1990), the observed $T_i$ equals $T_i(1)$ if $W_i = 1$, and $T_i(0)$ if $W_i = 0$ for each $i$. To proceed, we further require the ignorability and overlap assumptions, which are standard in causal survival analysis Kalbfleisch & Prentice (2002); Candès et al. (2023).

**Assumption 3.1** (Ignorability). $\{T(1), T(0)\} \perp\!\!\!\perp (W, C)|X$.

*Remark* 3.2. In addition to the treatment $W$, we assume that the potential outcome (*a.k.a*, true survival time) is also independent of the censoring time. This has been similarly assumed in Kalbfleisch & Prentice (2002) to achieve identifiability.

**Assumption 3.3** (Overlap). $0 < \mathbb{P}(W_i = 1|X) < 1$.

Given a nominal coverage level $\alpha \in (0, 1)$, our goal is to provide the lower predictive bound (LPB) $\hat{L}_{N,n}^{(w)}(X)$ for any treatment $w$, ensuring that it satisfies the marginal coverage.

$$\mathbb{P}_{X,T(w)}(T(w) \geq \hat{L}_{N,n}^{(w)}(X)) \geq 1 - \alpha,$$

where $N, n$ are the sample sizes of the training and calibration data.

To provide the LPB, previous works Gui et al. (2024); Davidov et al. (2025) introduced adaptive cut-off methods. These methods offered LPBs that achieve probably-approximately-correct (PAC)-type coverage, which is approximately marginal coverage based on the available data. However, their approaches may fail to achieve exact marginal coverage that considers the average across the whole population, including extreme cases that are omitted by PAC-type coverage, yet are crucial in survival analysis.

Specifically, Gui et al. (2024) considered the type-I censoring scenario, where we can observe $\{X_i, \widetilde{T}_i, C_i\}_i$. Given the estimated quantile regression function $\widehat{q}_\tau(X)$, they propose to search $\widehat{\tau}$ to meet the coverage guarantee, where $q_\tau$ is the true $\tau$-th quantile we want to estimate. First, they define $\alpha(\tau)$ such that

$$\begin{aligned}
\alpha(\tau) &:= \mathbb{P}(T \leq \widehat{q}_\tau(X)) \\
&\approx \frac{\mathbb{E}\left[1\{T < \widehat{q}_\tau(X) \leq C\}\widehat{\omega}_\tau(X)\right]}{\mathbb{E}\left[1\{\widehat{q}_\tau(X) \leq C\}\widehat{\omega}_\tau(X)\right]} \\
&\overset{(1)}{\approx} \frac{\sum_{i \in \mathcal{I}_{\text{cal}}} \widehat{w}_\tau(X_i)1\{T_i < \widehat{q}_\tau(X_i) \leq C_i\}}{\sum_{i \in \mathcal{I}_{\text{cal}}} \widehat{w}_\tau(X_i)1\{\widehat{q}_\tau(X_i) \leq C_i\}} \overset{\triangle}{=} \widehat{\alpha}(\tau),
\end{aligned}$$

where $\mathcal{I}_{\text{cal}}$ denotes the index set of calibration data, and $\widehat{w}_\tau(x)$ is chosen to be approximately equal to $1/\mathbb{P}(\widehat{q}_\tau(X) \leq C|X = x)$. Then, they adaptively choose a cut-off value $\widehat{\tau}$, such that $\widehat{\tau} := \sup\{\tau \in [0, 1] : \sup_{\tau' \leq \tau} \widehat{\alpha}(\tau') \leq \alpha\}$. Later, Davidov et al. (2025) extended this adaptive

---

**Algorithm 1** Conformal Survival Counterfactual Prediction.

---

**Input:** : Data $\mathcal{D} = \{W_i, X_i, \widetilde{T}_i, e_i\}$, counterfactual quantile regression estimator $\widehat{q}_\tau^{(w)}$ of $T(w)|X = x$ and function $\widehat{\omega}(x)$ to fit the weight function; level $\alpha$, testing point $x$.

1: Split the data into two folds: the training fold $\mathcal{D}_{\mathrm{tr}}$ and the calibration fold $\mathcal{D}_{\mathrm{cal}}$.
2: Define the non-conformity score function: $V_i^{(w)} := V^{(w)}(X_i, \widetilde{T}_i) = \widehat{q}_\tau^{(w)} - \widetilde{T}_i$.
3: Define $\mathcal{I}_{\mathrm{cal}}^{(w)} = \{i : (X_i, W_i, \widetilde{T}_i, e_i) \in \mathcal{D}_{\mathrm{cal}} \text{ with } W_i = w, e_i = 1\}$.
4: Compute the $V_i^{(w)}$ for each $i \in \mathcal{I}_{\mathrm{cal}}^{(w)}$.
5: Compute the weight $\widehat{\omega}(X_i) = 1/\widehat{\gamma}(X_i)$ for each $i \in \mathcal{I}_{\mathrm{cal}}^{(w)}$.
6: Compute the weights $\widehat{p}_i(x)$ and $\widehat{p}_\infty(x)$ from (2).
7: Compute the $c_{1-\alpha}^{(w)}(\tau) = \mathrm{Quantile}\left\{1 - \alpha; \sum_{i \in \mathcal{I}_{\mathrm{cal}}^{(w)}} \widehat{p}_i(x)\delta_{V_i^{(w)}} + \widehat{p}_\infty(x)\delta_\infty\right\}$

**Output:** : The calibrated Conformal Survival Counterfactual Prediction LPB: $\widehat{L}_{N,n}^{(w)}(X, \tau) = \widehat{q}_\tau^{(w)}(X) - c_{1-\alpha}^{(w)}(\tau)$.

---

cut-off to a general right-censored setting. However, this method can only achieve approximate marginal coverage due to "(1)" that the empirical average can approximate the population average with high probability. This motivates us to provide a new calibration procedure, which can achieve exact marginal coverage.

## 4 METHOD

In this section, we introduce our calibration procedure for survival counterfactual prediction. The core idea of our procedure lies in transforming the coverage probability into a reweighted expectation, and then applying a reweighting scheme for calibration. Section 4.1 introduces the procedure, and Section 4.2 provides the coverage guarantee.

### 4.1 CONFORMAL CALIBRATION

To begin, we split the dataset into a training data $\mathcal{D}_{\mathrm{tr}}$ and a holdout calibration data $\mathcal{D}_{\mathrm{cal}}$. Let $N = |\mathcal{D}_{\mathrm{tr}}|$, $n = |\mathcal{D}_{\mathrm{cal}}|$. Correspondingly, we use $\mathcal{I}_{\mathrm{tr}}$ and $\mathcal{I}_{\mathrm{cal}}$ to respectively denote the index set of the training and calibration data.

**Counterfactual quantile regression function.** Denote by $q_\tau^{(w)}(x)$ the true $\tau$-th quantile of $T(w)|X = x$ and $L_\alpha^{(w)}(x)$ the oracle LPB for the corresponding $\alpha$-th counterfactual conditional quantile function. Under the ignorability condition, we have

$$\mathbb{P}(T(w)|X = x) = \mathbb{P}(T(w)|X = x, W = w) = \mathbb{P}(T|X = x, W = w).$$

That means, the goal is to estimate the $\tau$-th quantile of $T|X = x, W = w$ from observed samples $\{\widetilde{T}_i, X_i, W_i\}$ with $\widetilde{T}_i := \min\{T_i, C_i\}$. We can then apply censored quantile regression (CQR) methods to estimate $q_\tau^{(w)}(x)$, including CQR Peng & Huang (2008), CQR forest Li & Bradic (2020), and CQR neural networks Pearce et al. (2022).

**Non-conformity score.** We follow Romano et al. (2019) to define the *non-conformity* score as $V_i^{(w)} := V^{(w)}(X_i, \widetilde{T}_i) = \widehat{q}_\tau^{(w)}(X_i) - \widetilde{T}_i$ on $\mathcal{D}_{\mathrm{cal}}$, indicating how atypical a value of the outcome is given observed covariate values, and a large value indicates a lack of conformity to training data. Let $c_{1-\alpha}^{(w)}(\tau)$ denotes the $1 - \alpha$ quantile of $V^{(w)}$, *i.e.*, $c_{1-\alpha}^{(w)}(\tau) := \inf_c\{\mathbb{P}(V^{(w)}(X_i, \widetilde{T} \leq c) \geq 1 - \alpha\}$.

**Calibration procedure.** Note that the adaptive cut-off method Gui et al. (2024) adaptively adjusted the $\tau$ such that $\alpha(\tau) := \mathbb{P}(T < \widehat{q}_\tau(X))$, because $\alpha(\tau)$ may not necessarily equal (or even close) to $\alpha$ when $\widehat{q}_\tau(X)$ does not estimate the $\tau$-th quantile of $T|X$ ($T|X, W = w$ in our scenario) well. Therefore, they used $\widehat{\alpha}(\tau)$, the empirical version of $\alpha(\tau)$ to learn the cut-off value. The gap between $\widehat{\alpha}(\tau)$ and $\alpha(\tau)$ makes their procedures achieve only approximate coverage.

To achieve the exact marginal coverage, instead of $\alpha(\tau) := \mathbb{P}(T < \widehat{q}_\tau(X))$, we provide the upper bound for $\mathbb{P}(V^{(w)}(X, \widetilde{T}) \leq c_{1-\alpha}^{(w)}(\tau))$, which is exactly equals to $\alpha$ and therefore avoid the need for

Table 1: Experiment results with different $\alpha$ within a reasonable range, with the corresponding optimized $\tau^*$, average coverage rate , and the corresponding optimal LPB. The results are evaluated on 10 independent trials on setting 4 with the same ratio as shown in Figure 1.

| $\alpha$ | 0.05 | 0.10 | 0.15 | 0.20 |
|---|---|---|---|---|
| $\tau^*$ | 0.16 | 0.16 | 0.26 | 0.21 |
| Average coverage rate | 0.958 | 0.914 | 0.872 | 0.845 |
| LPB with $\tau = \alpha$ | 0.411 | 0.778 | 1.19 | 1.57 |
| Optimal LPB | 0.503 | 0.803 | 1.25 | 1.64 |

adaptively adjustment. Specifically, we have:

$$
\begin{aligned}
\alpha &= \mathbb{P}(V^{(w)}(X, T(w)) \geq c_{1-\alpha}^{(w)}(\tau)) \\
&= \mathbb{P}(T(w) \leq \widehat{q}_\alpha^{(w)}(X) - c_{1-\alpha}^{(w)}(\tau)) \\
&= \mathbb{E}[\mathbb{P}(T(w) \leq \widehat{q}_\alpha^{(w)}(x) - c_{1-\alpha}^{(w)}(\tau)|X = x)] \\
&\overset{(i)}{=} \mathbb{E}_X[\mathbb{P}(T \leq \widehat{q}_\alpha^{(w)}(x) - c_{1-\alpha}^{(w)}(\tau)|X = x, W = w)] \\
&\overset{(ii)}{=} \mathbb{E}_X\left[\mathbb{P}(T \leq \widehat{q}_\alpha^{(w)}(x) - c_{1-\alpha}^{(w)}(\tau)|X = x, W = w)p(e = 1|x, W = w)\frac{1}{p(e = 1|x, W = w)}\right] \\
&\overset{(iii)}{\leq} \mathbb{E}_X\left[\mathbb{P}(T \leq \widehat{q}_\alpha^{(w)}(x) - c_{1-\alpha}^{(w)}(\tau), e = 1|X = x, W = w)\frac{1}{p(e = 1|x, W = w)}\right] \\
&\overset{(iv)}{=} \mathbb{E}_X\left[\mathbb{P}(T \leq \widehat{q}_\alpha^{(w)}(x) - c_{1-\alpha}^{(w)}(\tau)|e = 1, W = w, X = x\right] \\
&= \mathbb{E}\left[\mathbb{P}(T \leq \widehat{q}_\alpha^{(w)}(x) - c_{1-\alpha}^{(w)}(\tau), e = 1, W = w|X = x)\frac{1}{p(e = 1, W = w|x)}\right] \\
&\overset{\text{def}}{=} \mathbb{E}\left[\mathbb{I}\left(T \leq \widehat{q}_\alpha^{(w)}(x) - c_{1-\alpha}^{(w)}(\tau), W = w, e = 1\right)\frac{1}{\gamma(x)}\right] \\
&= \mathbb{E}\left[\mathbb{I}\left(V(\widetilde{T}, X) \geq c_{1-\alpha}^{(w)}(\tau)\right)\frac{p(W = w, e = 1)}{\gamma(x)}|W = w, e = 1\right],
\end{aligned}
\tag{1}
$$

where (i) follows from Assumption 3.1 and the SUTVA condition, (ii) comes from the tower property, (iii) is derived by the proof of Lemma A.1 conditional on $X = x, W = w$, and $\gamma(x) := p(W = w, e = 1|x)$. Through the upper bound in (iv), note that it is sufficient for the LPB $\widehat{L}_{N,n}^{(w)}(X, \tau)$ to satisfy the coverage guarantee for $\mathbb{P}_X \times \mathbb{P}_{\widetilde{T}|W=w,e=1,X}$ (since $T = \widetilde{T}$ given $e = 1$). Denote $\omega(x) = \frac{p(W=w,e=1)}{\gamma(x)}$. Since $\mathbb{E}_{X|W=w,e=1}[\omega(X)|W = w, e = 1] = 1$, we can employ the weighted conformal prediction Lei & Candès (2021) to ensure that

$$
\mathbb{E}\left[\mathbb{I}\left(V^{(w)}(\widetilde{T}, X) \geq c_{1-\alpha}^{(w)}(\tau)\right)\omega(X)|W = w, e = 1\right] \approx \alpha.
$$

To this end, for each $i \in \mathcal{I}_{\text{cal}}$ with $W = w$ and $e = 1$, we compute $\widehat{\omega}(X_i) = \frac{p(W=w,e=1)}{\widehat{\gamma}(X_i)}$. For the test data $x$, we compute $\widehat{\omega}(x) := \frac{p(W=w,e=1)}{\widehat{\gamma}(x)}$. Following Lei & Candès (2021), we can take

$$
c_{1-\alpha}^{(w)}(\tau) = \text{Quantile}\left\{1 - \alpha; \sum_{i \in \mathcal{I}_2^{(w)}} \widehat{p}_i(x)\delta_{V_i^{(w)}} + \widehat{p}_\infty(x)\delta_\infty\right\},
$$

where

$$
\widehat{p}_i(x) = \frac{\widehat{\omega}(X = x_i)}{\sum_{i \in \mathcal{I}_{\text{cal}}^{(w)}} \widehat{\omega}(X = x_i) + \widehat{\omega}(x)}, \quad \widehat{p}_\infty(x) = \frac{\widehat{\omega}(x)}{\sum_{i \in \mathcal{I}_{\text{cal}}^{(w)}} \widehat{\omega}(X = x_i) + \widehat{\omega}(x)}
\tag{2}
$$

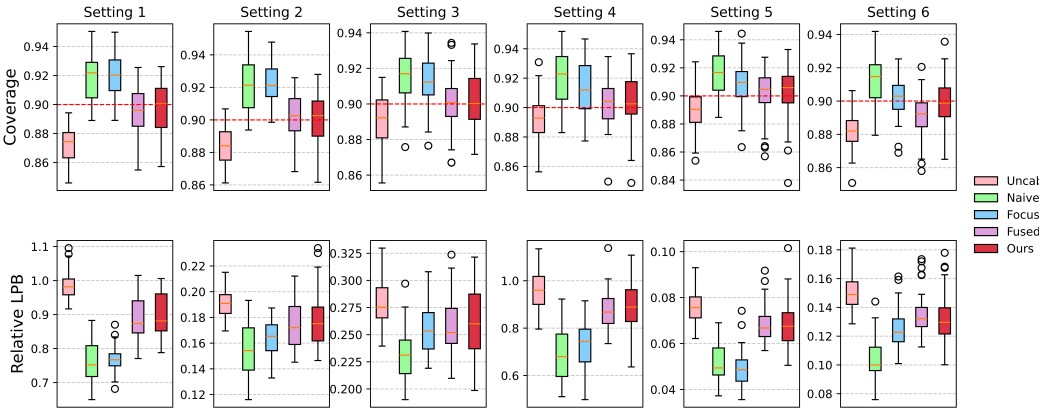

Figure 1: Comparison of the different calibration methods in different settings on the synthetic datasets. **Top:** empirical coverage rate, with a red dashed line indicating the nominal 90% level. **Bottom:** relative LPB obtained by different methods. A higher relative LPB is better. The results are evaluated on 50 independent trials, each consisting of newly sampled train, validation, calibration, and test sets with ratios of 50%, 10%, 30%, and 10% of the synthetic datasets, respectively, totaling 3000 samples.

Since $p(W = w, e = 1)$ can be canceled out, it is sufficient to only estimate $\widehat{\gamma}(x)$. Therefore, with a bit of abuse of notation, we denote $\widehat{\omega}(x) := \frac{1}{\widehat{\gamma}(x)}$. The LPB can be then given by: $\widehat{L}_{N,n}^{(w)}(X, \tau) := \widehat{q}_{\tau}^{(w)}(X) - c_{1-\alpha}^{(w)}(\tau)$. The whole procedure is summarized in Algorithm 1.

**LPB optimization.** As we will demonstrate in the next subsection, our procedure yields a prediction set that satisfies the coverage guarantee for any $\tau \in (0, 1)$. To ensure the LPB is as informative as possible, we choose $\tau$ to maximize $\widehat{L}_{N,n}^{(w)}(X, \tau)$ for any test data $X$. Specifically, for each $X = x$, we obtain:

$$\tau^*(x) := \arg \max_{\tau \in (0,1)} \left( \widehat{q}_{\tau}^{(w)}(x) - c_{1-\alpha}^{(w)}(\tau)(x) \right),$$

The optimized LPB on the test data $x$ is given by $\widehat{L}_{N,n}^{(w)}(X, \tau^*(x)) = \widehat{q}_{\tau^*}^{(w)}(x) - c_{1-\alpha}^{(w)}(\tau)(x)$.

## 4.2 THEORETICAL GUARANTEE

Our theoretical analysis builds on Lei & Candès (2021); Candès et al. (2023). Specifically, by (iv) in (1), the problem reduces to constructing the LPB for the distribution $\mathbb{P}_X \times \mathbb{P}_{\widetilde{T}|W=w,e=1,X}$, from the data

$$\left\{ X_i, \widetilde{T}_i \right\}_{i \in \mathcal{I}_{\mathrm{cal}}, W_i = w, e_i = 1} \sim \mathbb{P}_{X|W=w,e=1} \times \mathbb{P}_{\widetilde{T}|W=w,e=1,X}.$$

To achieve calibration, the weight function $\omega(x)$ is introduced such that

$$\omega(x) = \frac{d\mathbb{P}_X}{d\mathbb{P}_{X|W=w,e=1}}(x) = \frac{p(W = w, e = 1)}{\gamma(x)}. \tag{3}$$

We then propose to estimate $\widehat{\omega}(x)$ from the training data. Theorem 4.1 establishes a distribution-free exact guarantee for counterfactual prediction intervals under covariate shift, in which $\widehat{\omega}(x)$ is the estimated density ratio quantifying distribution shift via Radon-Nikodym derivative $(d\mathbb{P}_X/d\mathbb{P}_{X|W=w,e=1})(x)$.

**Theorem 4.1.** *Let* $(X_i, \widetilde{T}_i)_{i \in \mathcal{I}_{\mathrm{cal}}, W_i = w, e_i = 1} \overset{i.i.d}{\sim} \mathbb{P}_{X|W=w,e=1} \times \mathbb{P}_{T(w)|X,e=1}$. *Set* $N = |\mathcal{D}_{\mathrm{tr}}|$ *and* $n = |\mathcal{D}_{\mathrm{cal}}|$. *Further, let* $\widehat{q}_{\alpha}^{(w)}(x) = \widehat{q}_{\alpha}^{(w)}(x; \mathcal{D}_{\mathrm{tr}})$ *be an estimate of the* $\alpha$-*th conditional quantile* $q_{\alpha}^{(w)}(x)$ *of* $T(w)|X = x$, $\widehat{\omega}(x) = \widehat{\omega}(x; \mathcal{D}_{\mathrm{tr}})$ *be an estimate of* $\omega(x) = (d\mathbb{P}_X/d\mathbb{P}_{X|W=w,e=1})(x)$, *and* $\widehat{L}_{N,n}^{(w)}(x)$ *be the counterfactual LPB resulting from Algorithm 1. Assume that* $\mathbb{E}[\widehat{\omega}|\mathcal{D}_{\mathrm{tr}}] < \infty$, *where* $\mathbb{E}$ *denotes expectation over* $X \sim \mathbb{P}_{X|W=w,e=1}$. *Redefine* $\widehat{\omega}(x)$ *as* $\widehat{\omega}(x)/\mathbb{E}[\widehat{\omega}(x)|\mathcal{D}_{\mathrm{tr}}]$ *so that* $\mathbb{E}[\widehat{\omega}(X)|\mathcal{D}_{\mathrm{tr}}] = 1$. *Then we have:*

$$\mathbb{P}_{(X,T(w)) \sim \mathbb{P}_X \times \mathbb{P}_{T(w)|X,e=1}} \left( T(w) \geq \widehat{L}_{N,n}^{(w)}(X) \right) \geq 1 - \alpha - \frac{1}{2}\mathbb{E}_{X \sim \mathbb{P}_{X|W=w,e=1}}\big[ |\widehat{\omega}(X) - \omega(X)| \big]. \tag{4}$$

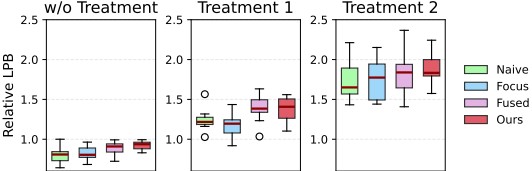

Figure 2: LPB comparison from the methods with coverage guarantee of $\alpha = 0.1$ for multi-treatment regimes with the same ratios to each other on setting 4, with 10 independent trials.

This bound quantifies how estimation error in the density ratio $(\widehat{\omega}_N(x) - \omega(x))$ affects the coverage probability, with the error term diminishing as the density ratio estimation improves. The weight estimator is normalized to ensure $\mathbb{E}[\widehat{\omega}(X)|\mathcal{D}_{\mathrm{tr}}] = 1$, which stabilizes the variance and enables tractable theoretical analysis. The result holds without strong distributional assumptions, maintaining the non-parametric spirit of conformal methods while extending them to counterfactual settings.

The following theorem provides the doubly robust property for counterfactual prediction intervals. This theoretical guarantee ensures that our method maintains valid coverage even when either the weights function $\widehat{\gamma}(x)$ or the counterfactual quantile estimator $\widehat{q}^{(w)}(x)$ is misspecified, provided one of them is consistently estimated. The doubly robust mechanism provides mutual compensation between the two estimation approaches: when the weights function is inaccurate, the quantile estimation compensates through Assumption **A**1, and vice versa, when the quantile estimation is misspecified, the weights function provides robustness through Assumption **A**2.

**Theorem 4.2.** *Let $N = |\mathcal{D}_{\mathrm{tr}}|$, $n = |\mathcal{D}_{\mathrm{cal}}|$, and $\widehat{q}_\alpha^{(w)}(x) = \widehat{q}_\alpha^{(w)}(x|\mathcal{D}_{\mathrm{tr}})$ denote the estimate of $\alpha$-th conditional quantile $q_\alpha^{(w)}(x)$ of $T(w)|X = x$. Further, let $\widehat{\gamma}(x) = \widehat{\gamma}(x|\mathcal{D}_{\mathrm{tr}})$ denote the estimate of $\gamma(x)$, and $\widehat{L}_{N,n}^{(w)}(x)$ denote the corresponding counterfactual LPB resulting from Algorithm 1. Assume that $\mathbb{E}[1/\widehat{\gamma}(X)|\mathcal{D}_{\mathrm{tr}}] < \infty$ and $\mathbb{E}[1/\gamma(X)] < \infty$. Assume that one of the following holds:*
*A1* $\lim_{N \to \infty} \mathbb{E}\left[\left|\frac{1}{\widehat{\gamma}(x)} - \frac{1}{\gamma(x)}\right|\right] = 0$;
*A2* *(i) there exists $r, b_1, b_2 > 0$ such that $\mathbb{P}(T(w) = t|X = x) \in [b_1, b_2]$ uniformly over all $(x, t)$ with $t \in [q_\alpha^{(w)}(x) - r, q_\alpha^{(w)}(x) + r]$,*

*(ii) let $\mathcal{E}_N(X) = |\widehat{q}_{\beta,N}^{(w)}(x) - q_\beta^{(w)}(x)|$, there exist $\delta > 0$ such that*

$$\mathbb{E}\left[\frac{1}{\widehat{\gamma}(x)^{1+\delta}}|\mathcal{D}_{\mathrm{tr}}\right] < \infty, \quad \lim_{N \to \infty}\left[\frac{\mathcal{E}_N(X)}{\widehat{\gamma}_N(x)}\right] = \lim_{N \to \infty}\left[\frac{\mathcal{E}_N(X)}{\gamma(x)}\right]. \tag{5}$$

*Then under SUTVA and the strong ignorability assumption,*

$$\lim_{N,n \to \infty} \mathbb{P}_{(X,T(w)) \sim \mathbb{P}_X \times \mathbb{P}_{T(w)|X,e=1}}\left(T(w) \geq \widehat{L}_{N,n}^{(w)}(x)\right) \geq 1 - \alpha. \tag{6}$$

*Furthermore, if **A2** holds, then for any $\epsilon > 0$,*

$$\lim_{N,n \to \infty} \mathbb{P}_{X \sim \mathbb{P}_{X|W=w,e=1}}\left(\mathbb{P}(T(w) \geq \widehat{L}_{N,n}^{(w)}(x)|X) \leq 1 - \alpha - \epsilon\right) = 0 \tag{7}$$

Theorem 4.2 is a special case of Corollary B.4 in Appendix B. It is easy to see that Theorem 4.2 applies to counterfactual predictions across all treatments. Moreover, property (7) implies that conformal survival counterfactual analysis has approximately guaranteed conditional coverage for counterfactuals if the conditional quantiles are estimated well.

## 5 EXPERIMENT

We evaluate both the coverage rate and the average LPB over the test set on synthetic data and real clinical data, with the desired coverage level $1-\alpha$ to 90%. In Section 5.1, we compare different calibration procedures on different scenarios with simulation for censoring and treatment rates in real-world clinical trials. We further validate our method using some real clinical cases collected from a cancer hospital in Section 5.2.

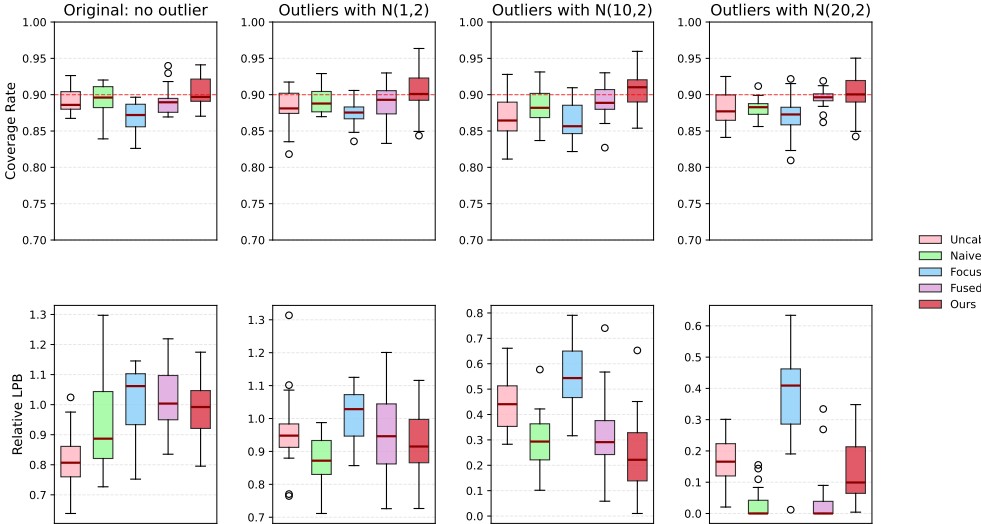

Figure 3: The outliers of survival time are simulated by subtracting the corresponding normal noise for 10% data randomly selected. The simulated outliers, with increasingly smaller values from the left to the right column, with details listed in Appendix D.2.

## 5.1 SIMULATION

**Data.** Simulation is crucial for verifying the efficiency of performance, as the survival time $T$ is often lacking. We test our method on different settings designed to mimic the censorship in real-world clinical trials, as in Candès et al. (2023); Gui et al. (2024); Davidov et al. (2025). With the details described in Appendix C.1.

**Models.** The model utilized is a multilayer perceptron (MLP) with only one hidden layer to reduce the potential overfitting, implemented in PyTorch. And we fit Random Forest classifiers to estimate the weights function $\omega(x)$. For more details, please refer to Appendix D.

**Results.** All methods aim to achieve a coverage rate of $1 - \alpha = 90\%$, corresponding to the red dashed line shown in Figure 1. The larger the relative LPB, the more informative it is. The results of synthetic data are shown in Figure 1. In comparison, we evaluate our method against the *uncab* method applied without calibration, the *naive* calibrated method, the *focused* calibration method Davidov et al. (2025), and the *fused* calibration method in Davidov et al. (2025).

Our method consistently achieves a more informative median-value LPB while ensuring it is closest to the desired coverage rates across all six experimental settings. Although the average coverage rate of our method slightly falls below $1 - \alpha$ in setting 6, it remains remarkably close to the target level. It simultaneously demonstrates the highest LPB among all methods that guarantee the desired coverage rates (including naive and focused methods). In settings 3, 4, and 5, our method achieves valid coverage rates comparable to the fused approach while yielding significantly larger LPB values. Although the resulting prediction intervals are wider, our method provides exact statistical guarantees for the coverage probability.

Our method also achieves desired coverage with outliers. To verify the robustness of our method, we introduce outlier data (details in Appendix D.2) into Setting 4 and report the resulting coverage rates and LPB in Figure 3. The results show that our method consistently maintains the desired coverage guarantee, whereas the compared baselines fail to do so. In particular, those methods-"Focus" and "Fused" Davidov et al. (2025)-with PAC-type guarantee do not necessarily achieve the marginal coverage in the presence of outlier data.

**Additional Experiment.** Besides the comparative experiments above demonstrating the effectiveness and robustness of our method, some additional experiments are conducted in Appendix E. We first verify in Appendix E.1 that once the sample sizes reach a certain threshold, our method can stably achieve the coverage guarantee while producing less conservative LPB. In Appendix E.2, we further validate that the LPB of our method effectively captures adaptiveness to different covariates.

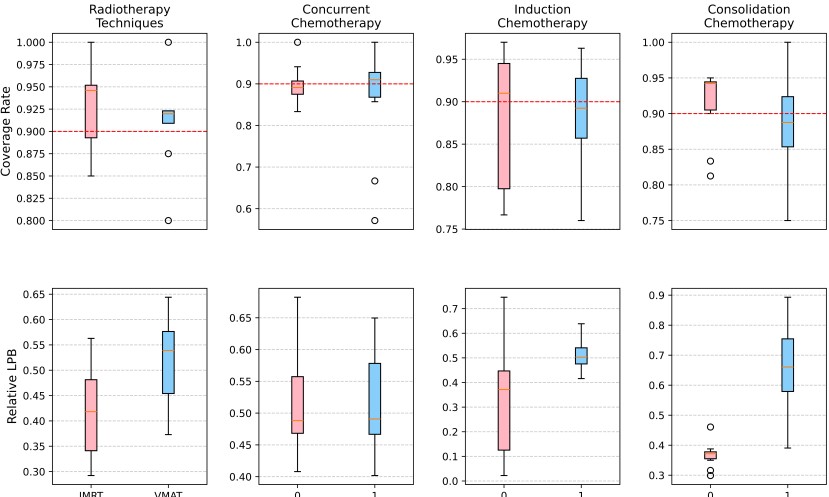

Figure 4: Performance in real data of different kinds of radiochemotherapy regimens: radiotherapy technique, concurrent chemotherapy, induction chemotherapy, and consolidation chemotherapy. **Top:** empirical coverage rate, with a red dashed line indicating the nominal 90% level. **Bottom:** LPB (years) obtained by our method. A higher relative LPB indicates the corresponding treatment is better. The results are evaluated on 10 independent trials, each consisting of newly split train, validation, calibration, and test sets with ratios of 50%, 10%, 30%, and 10% of the clinical dataset. The ratios of the four above radiochemotherapies are listed in Table 4.

In Appendix E.3, we explore the impact of $p(W = w, e = 1)$ on our method, with results consistent with the expectations derived from (3). Besides, Appendix E.4 and Appendix E.5 present sensitivity analysis with respect to the regression algorithm and weight function, respectively. The results show that our procedure consistently attains the desired coverage. In addition, we report the value of $\tau^*$ selected by our optimization procedure, along with the corresponding coverage rate and LPB in Table 1 and Figure 11. As shown in Figure 11, the LPB achieved at $\tau^*$ is comparable to that at $\tau = \alpha$, indicating that the quantile regression model is well trained. Finally, we expand setting 4 to a multi-treatment scenario. Figure 2 shows that the LPB varies across treatments but consistently satisfies the coverage guarantee. Please refer to Appendix D.1 for additional details.

## 5.2 APPLICATION ON REAL DATA

**Data.** We evaluate our method on a real-world dataset of 541 non-small cell lung cancer patients from a cancer hospital. Four different radiochemotherapy regimens in real data are examined, and the details of radiochemotherapy and the proportion of patients receiving different treatments is shown in Table 4. The dataset includes 124 clinical and quantitative radiomic features of lung cancer. Detail of the dataset is provided in Appendix C.2.

**Models.** The model trained for real data is a multilayer perceptron (MLP) with three hidden layers, implemented in PyTorch. And we also fit Random Forest classifiers to estimate the weights function $\omega(x)$. For more details, please refer to Appendix D.

**Results.** To validate our method's performance and reliability in reality, we apply it to a real-world dataset stratified by different treatment regimens. First, we investigate the effects of four distinct radiochemotherapy regimens on survival time. As shown in the Figure 4, the result shows a higher median LPB than those treated under intensity modulated radiation therapy (IMRT), which is consistent with the VMAT's better clinical benefits in lung cancer Hunte et al. (2022). Besides, as for the three chemotherapies, the addition of induction chemotherapy and concurrent chemotherapy shows higher LPBs, consistent with prior studies Curran et al.; Aguado et al. (2022). Our analysis also reveals a higher LPB under consolidation chemotherapy, potentially due to more favorable baseline characteristics Liu et al..

Then, we explore the adaptiveness of the LPB on cases under the VMAT technique. We select a set of covariates from three categories of known prognostic factors for survival time, including three

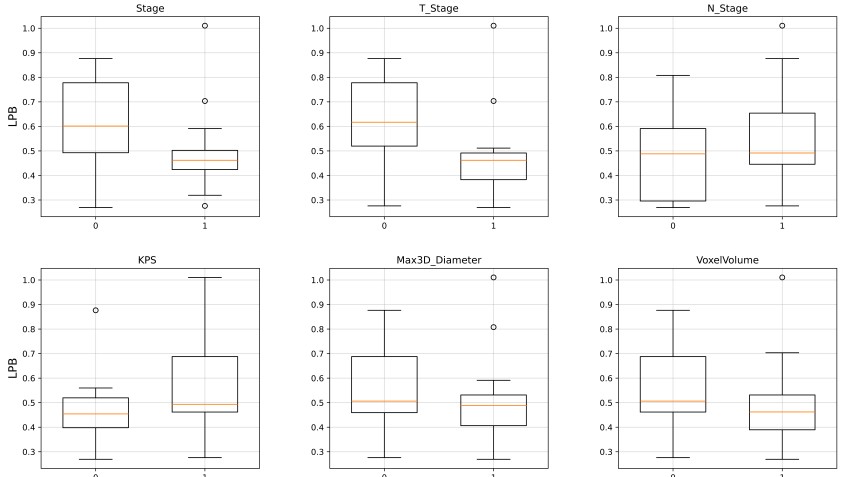

Figure 5: Adaptiveness analysis on clinical dataset of the stage, T-Stage, N-Stage, KPS, Max3D-Diameter, and Voxel-Volume. A higher LPB indicates that cases within the corresponding variable subgroup have longer survival times when receiving the current treatment. The details of the above six covariates are described in Table 5.

tumor-related clinical factors (i.e., overall stage, T-stage, and N-stage), one host-related clinical factor (i.e., Karnofsky Performance Status (KPS)), and two quantitative radiomic features of the tumor (i.e., Max3D-Diameter, Voxel-Volume). To better assess how these covariates influence LPB, all covariates are binarized, as listed in Table 5. As shown in the Figure 5, patients with more advanced stages, larger quantitative radiomic features tend to have larger tumor burdens and greater lymph node involvement Aerts et al. (2014); Amin et al. (2017), and shorter survival time. The patients with better body performance status (i.e., larger KPS value) usually have better functional status with longer survival time Quinten et al. (2009). Additionally, Appendix E.6 implements baseline methods that satisfy coverage guarantees in the simulation settings, and reports the corresponding results in Figure 10. As shown, our method can produce higher LPB, suggesting the informativeness of our procedure.

The LPB of our method in real data consistently correlates with important factors while maintaining the desired coverage rate, which demonstrates the applicability of our approach in complex, heterogeneous real-world scenarios and the potential in supporting personalized clinical decision-making for tumor treatment.

# 6 DISCUSSION

We introduce an uncertainty quantification procedure of counterfactuals with exact LPB coverage guarantees for general right-censored survival data. Under the SUTVA and strong ignorability assumptions, we achieve counterfactual prediction for a new test point by training a counterfactual quantile regressor and performing a weighted adjustment of the non-conformity score using the counterfactual calibration set. Although the assumptions employed in this paper are commonly used in causal inference, it isn't easy to guarantee that all these assumptions hold completely during real-world data collection processes. Therefore, in practical applications, appropriately constraining data that violate these assumptions in real datasets could enhance the robustness of our method in counterfactual prediction Oliveira et al. (2024); Feldman & Romano (2024). In actual follow-up studies, extreme scenarios such as imbalanced treatment usage proportions and high censoring rates may lead to inaccurate estimation of $\gamma(x)$, consequently increasing the uncertainty in counterfactual outcome predictions under such treatments. Hence, mitigating the impact of data imbalance is crucial Gui et al. (2024). Furthermore, beyond demonstrating differences between treatments, accurate quantitative estimation of the causal effects between different treatment outcomes holds significant importance for decision-making Lopez & Gutman (2017); Hu & Gu (2021); Lei & Candès (2021).

ACKNOWLEDGEMENT

This work was supported in part by the State Key Program of National Natural Science Foundation of China (Grant No.12331009), Young Scientists Fund of the National Natural Science Foundation of China (Grant No.KRH2305058), Funded by Tianjin Key Medical Discipline Construction Project (Grant No.TJYXZDXK-3-004B), and National Natural Science Foundation of China (No.82303672, No.82573437).

ETHICS STATEMENT

The institutional ethics committee approves the usage of the clinical dataset, and all the cases are anonymized in this work. No specific information is provided to preserve anonymity.

REPRODUCIBILITY STATEMENT

The proof of Theorem 4.1 and Theorem 4.2 is provided in Appendix A.1 and Appendix B.3 respectively. The generation details of synthetic data are provided in Appendix C.1. The model parameters and implementation are provided in Appendix D. Our code is available at conformalized survival counterfactual code.

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

## A PROOF OF THEORETICAL GUARANTEE

**Lemma A.1.** *Suppose that $x \in \mathcal{X}$, $\widehat{q}^{(w)} : \mathcal{X} \to \mathbb{R}^+$. Under Assumption 3.1 and if $\mathbb{P}(T \le \widehat{q}_\alpha^{(w)}(x) - c_{1-\alpha}^{(w)}|X = x, W = w) > 0$, then we have:*

$$\mathbb{P}(T \le \widehat{q}_\alpha^{(w)}(x) - c_{1-\alpha}^{(w)}|X = x, W = w)\mathbb{P}(e = 1|x, W = w)) \le \mathbb{P}(T \le \widehat{q}_\alpha^{(w)}(x) - c_{1-\alpha}^{(w)}, e = 1|X = x, W = w),$$
(8)

*where the probability is taken over $\mathbb{P}_{(W,X,T,e)}$*

*Proof.* Suppose $x \in \mathcal{X}$, All derivations in this proof are conditional on $X = x, W = w$, and so we denote $\mathbb{P}_{x,(w)}(\cdot) = \mathbb{P}(\cdot|X = x, W = w)$ and $\widehat{L}_\alpha = \widehat{q}_\alpha^{(w)}(x) - c_{1-\alpha}^{(w)}$ here for simplicity.

We begin by developing $\mathbb{P}(T \le C|T \le \widehat{L}_\alpha)$:

$$\begin{aligned}
\mathbb{P}_{x,(w)}(T < C \mid T < \widehat{L}_\alpha) &= \mathbb{P}_{x,(w)}(T < C \mid T < \widehat{L}_\alpha, C \ge \widehat{L}_\alpha)\mathbb{P}_{x,(w)}(C \ge \widehat{L}_\alpha) \\
&\quad + \mathbb{P}_{x,(w)}(T < C \mid T < \widehat{L}_\alpha, C < \widehat{L}_\alpha)\mathbb{P}_{x,(w)}(C < \widehat{L}_\alpha) \\
&= 1 \cdot \mathbb{P}_{x,(w)}(C \ge \widehat{L}_\alpha) \\
&\quad + \frac{\mathbb{P}_{x,(w)}(T < C, T < \widehat{L}_\alpha, C < \widehat{L}_\alpha)}{\mathbb{P}_{x,(w)}(T < \widehat{L}_\alpha, C < \widehat{L}_\alpha)}\mathbb{P}_{x,(w)}(C < \widehat{L}_\alpha) \\
&= \mathbb{P}_{x,(w)}(C \ge \widehat{L}_\alpha) \\
&\quad + \frac{\mathbb{P}_{x,(w)}(T < C, T < \widehat{L}_\alpha, C < \widehat{L}_\alpha)}{\mathbb{P}_{x,(w)}(T < \widehat{L}_\alpha) \cdot \mathbb{P}_{x,(w)}(C < \widehat{L}_\alpha)}\mathbb{P}_{x,(w)}(C < \widehat{L}_\alpha) \\
&= \mathbb{P}_{x,(w)}(C \ge \widehat{L}_\alpha) + \frac{\mathbb{P}_{x,(w)}(T < C, T < \widehat{L}_\alpha, C < \widehat{L}_\alpha)}{\mathbb{P}_{x,(w)}(T < \widehat{L}_\alpha)} \\
&= \mathbb{P}_{x,(w)}(C \ge \widehat{L}_\alpha) + \frac{\mathbb{P}_{x,(w)}(T < C, C < \widehat{L}_\alpha)}{\mathbb{P}_{x,(w)}(T < \widehat{L}_\alpha)},
\end{aligned}$$
(9)

where the first transition is by the law of total probability and the third is by Assumption 3.1. Then we develop $\mathbb{P}_{x,(w)}(T < C)$:

$$\mathbb{P}_{x,(w)}(T < C) = \mathbb{P}_{x,(w)}(T < C, C < \widehat{L}_\alpha) + \mathbb{P}_{x,(w)}(T < C, C \ge \widehat{L}_\alpha) \tag{10}$$

By subtraction (9) and (10), then we get:

$$\begin{aligned}
\mathbb{P}_{x,(w)}(T < C \mid T < \widehat{L}_\alpha) - \mathbb{P}_{x,(w)}(T < C) &= \mathbb{P}_{x,(w)}(C \ge \widehat{L}_\alpha) + \frac{\mathbb{P}_{x,(w)}(T < C, C < \widehat{L}_\alpha)}{\mathbb{P}_{x,(w)}(T < \widehat{L}_\alpha)} \\
&\quad - (\mathbb{P}_{x,(w)}(T < C, C < \widehat{L}_\alpha) + \mathbb{P}_{x,(w)}(T < C, C \ge \widehat{L}_\alpha)) \\
&= \mathbb{P}_{x,(w)}(C \ge \widehat{L}_\alpha) - \mathbb{P}_{x,(w)}(T < C, C \ge \widehat{L}_\alpha) \\
&\quad + \left(\frac{\mathbb{P}_{x,(w)}(T < C, C < \widehat{L}_\alpha)}{\mathbb{P}_{x,(w)}(T < \widehat{L}_\alpha)} - \mathbb{P}_{x,(w)}(T < C, C < \widehat{L}_\alpha)\right) \\
&= \mathbb{P}_{x,(w)}(T \ge C, C \ge \widehat{L}_\alpha) \\
&\quad + \left(\frac{1 - \mathbb{P}_{x,(w)}(T < \widehat{L}_\alpha)}{\mathbb{P}_{x,(w)}(T < \widehat{L}_\alpha)}\right)\mathbb{P}_{x,(w)}(T < C, C < \widehat{L}_\alpha) \\
&\ge 0,
\end{aligned}$$
(11)

then, multiply (11) by $\mathbb{P}_{x,(w)}(T < \widehat{L}_\alpha)$ we have that:

$$\mathbb{P}_{x,(w)}(T < C, T < \widehat{L}_\alpha) - \mathbb{P}_{x,(w)}(T < C)\mathbb{P}_{x,(w)}(T < \widehat{L}_\alpha) \ge 0. \tag{12}$$

We finish the proof with replacing $(T < C, T < \widehat{L}_\alpha)$ in (12) by $(e = 1, T \le \widehat{q}_\alpha^{(w)}(x) - c_{1-\alpha}^{(w)})$.

$\square$

Let Quantile$(\beta, F)$ denote the $\beta$-th quantile of a distribution function $F$, i.e.

$$\text{Quantile}(\beta, F) = \inf\{z : F(z) \geq \beta\} = \sup\{z : F(z) < \beta\}.$$

**Lemma A.2.** *(Equation (2) in Lemma 1 from Tibshirani et al. (2019) ). Let $v_1, \ldots, v_{n+1} \in \mathbb{R}$, and $(p_1, \ldots, p_{n+1}) \in \mathbb{R}$ be non-negative reals summing to 1. Then for any $\beta \in [0, 1]$ and*

$$v_{n+1} \leq \text{Quantile}\left(\beta, \sum_{i=1}^{n+1} p_i \delta_{v_i}\right) \iff v_{n+1} \leq \text{Quantile}\left(\beta, \sum_{i=1}^{n} p_i \delta_{v_i} + p_{n+1}\delta_\infty\right).$$

*Proof.* Consider the following useful fact about quantiles of a discrete distribution $F$, with support point $a_1, \ldots, a_k \in \mathbb{R}$: denoting $q = \text{Quantile}(\beta; F)$, if we reassign the points $a_i > q$ to arbitrary values strictly larger than $q$, yielding a new distribution $\widetilde{F}$, then the level $\beta$ quantile remains unchanged, Quantile$(\beta; F) = \text{Quantile}(\beta; \widetilde{F})$). Using this fact,

$$V_{n+1} > \text{Quantile}(\beta; V_{1:n} \cup \{\infty\}) \iff V_{n+1} > \text{Quantile}(\beta; V_{1:(n+1)}),$$

equivalently, we have that

$$V_{n+1} \leq \text{Quantile}(\beta; V_{1:n} \cup \{\infty\}) \iff V_{n+1} \leq \text{Quantile}(\beta; V_{1:(n+1)}),$$

$\square$

**Lemma A.3.** *(Equation (10) from Berrett et al. (2020)). Let $d_{\text{TV}}(\mathbb{P}_{1X}, \mathbb{P}_{2X})$ denote the total-variation distance between distributions $P_{1X}$ and $P_{2X}$. Then*

$$d_{\text{TV}}(P_{1X} \times P_{Y|X}, P_{2X} \times P_{Y|X}) = d_{\text{TV}}(P_{1X}, P_{2X}).$$

*Proof.* For any $(P, Q)$ and $(P', Q')$, if $(P|Q = y) \overset{d}{=} (P'|Q' = y)$ for any $y$, then by the definition of total-variation we have:

$$\begin{aligned}
d_{\text{TV}}((P, Q), (P', Q)) &= \frac{1}{2} \int |p(x)q(y) - p'(x)q(y)| dx dy \\
&= \frac{1}{2} \int q(y) dy \, |p(x) - p'(x)| dx \\
&= \frac{1}{2} \int |p(x) - p'(x)| dx \\
&= d_{\text{TV}}(P, P')
\end{aligned}$$

$\square$

### A.1 PROOF OF THEOREM 4.1

*Proof.* **Step I.** Under $\mathbb{P}_{X \sim \mathbb{P}_X}(w(X) < \infty) = 1$. We first consider the case where $\mathbb{P}_X$ is absolutely continuous with respect to $\mathbb{P}_{X|W=w, e=1}$, i.e.

$$\mathbb{P}_{X \sim \mathbb{P}_X}(\omega(X) < \infty) = 1.$$

In this case, for any measurable function $f$,

$$\mathbb{E}_{X \sim \mathbb{P}_X}[f(X)] = \mathbb{E}_{X \sim \mathbb{P}_{X|W=w, e=1}}[\omega(X)f(X)]. \tag{13}$$

On the other hand, it always holds that $\mathbb{P}_{X \sim \mathbb{P}_{X|W=w, e=1}}(\omega(X) < \infty) = 1$. In addition, the assumption $\mathbb{E}_{X \sim \mathbb{P}_{X|W=w, e=1}}[\widehat{\omega}(X)|\mathcal{Z}_{\text{tr}}]$ implies that $\mathbb{P}_{X \sim \mathbb{P}_{X|W=w, e=1}}(\widehat{\omega}(X) < \infty) = 1$. By (13), we have

$$\mathbb{P}_{X \sim \mathbb{P}_X}(\widehat{\omega}(X) < \infty) = 1 - \mathbb{E}_{X \sim \mathbb{P}_{X|W=w, e=1}}[\omega(X)I(\widehat{\omega}(X) = \infty)].$$

Since the integrand is non-negative,

$$\mathbb{E}_{X \sim \mathbb{P}_{X|W=w, e=1}}[\omega(X)I(\widehat{\omega}(X) = \infty)] = \lim_{k \to \infty} \mathbb{E}_{X \sim \mathbb{P}_{X|W=w, e=1}}[\omega(X)I(\omega(X) \leq K, \widehat{\omega}(X) = \infty)] \tag{14}$$

$$\leq \lim_{k \to \infty} K \mathbb{E}_{X \sim \mathbb{P}_{X|W=w, e=1}}[\widehat{\omega}(X) = \infty].$$

Thus, we also have

$$\mathbb{P}_{X \sim \mathbb{P}_X}(\widehat{\omega}(X) < \infty) = 1.$$

Index the counterfactual calibration fold $\mathcal{D}_{\text{cal}}^{(w)} := \{(X_i, W_i, \tilde{T}_i, e_i) \in \mathcal{D}_{cal} \text{ with } W_i = w, e_i = 1\}$ by $\{1, \ldots, n\}$ and let $(X_{n+1}, T_{n+1}(w)) \sim \mathbb{P}_X \times \mathbb{P}_{T(w)|X,e=1}$. Write $Z_i^{(w)}$ for $(X_i, T_i(w)) \in \mathcal{D}_{\text{cal}}^{(w)}$ under the SUTVA assumption, and $V^{(w)}$ for $(V_1^{(w)}, \ldots, V_{n+1}^{(w)})$. For notational convenience, we suppress the subscripts $N$ and $n$ in $\widehat{q}; \widehat{\omega}; \widehat{L}^{(w)}$ as well as in $\widehat{p}_i(x)$ and $c_{1-\alpha}^{(w)}(x)$. Next, for any permutation $\pi$ on $\{1, \cdots, n+1\}$ and $v^{(w)*} \in \mathbb{R}^{n+1}$, let $v_\pi^{(w)*} = (v_{\pi(1)}^{(w)*}, \cdots, v_{\pi(n+1)}^{(w)*})$. Further, let $\mathscr{L}(z^{(w)})$ be the joint density of $\mathcal{Z}^{(w)} = (Z_1^{(w)}, \cdots, Z_{n+1}^{(w)})$ and $p(z^{(w)})$ be the density of $Z^{(w)}$ (with respect to a dominating measure). Letting $\mathcal{E}(v^{(w)})$ denote the unordered set of $v^{(w)}$, it is easy to see that

$$(V^{(w)}|\mathcal{E}(V^{(w)})) = (\mathcal{E}(v^{(w)*}), \mathcal{D}_{\text{tr}}) \stackrel{d}{=} v_\Pi^{(w)*}, \tag{15}$$

where $\Pi$ is a random permutation with

$$\mathbb{P}(\Pi = \pi | \mathcal{D}_{\text{tr}}) = \frac{\mathscr{L}(z_\pi^{(w)*})}{\sum_\pi \mathscr{L}(z_\pi^{(w)*})} = \frac{\omega(X_{\pi(n+1)})}{\sum_\pi \omega(X_{\pi(n+1)})} = \frac{\omega(X_{\pi(n+1)})}{n! \sum_{i=1}^{n+1} \omega(X_i)}.$$

Note that this conditional probability is well-defined because $\omega(X) < \infty$ almost surely under both $\mathbb{P}_{X|W=w,e=1}$ and $\mathbb{P}_X$. As a result, for any $j \in \{1, 2, \cdots, n+1\}$,

$$\mathbb{P}(\Pi(n+1) = j | \mathcal{D}_{\text{tr}}) = \frac{\omega(X_j)}{\sum_{i=1}^{n+1} \omega(X_i)} = p_j(X_{n+1}),$$

where $p_{n+1}$ denotes $p_\infty$ for notational convenience. With (15), this gives

$$(V_{n+1}^{(w)}|\mathcal{E}(V^{(w)}) = \mathcal{E}(v^{(w)*}), \mathcal{D}_{\text{tr}}) \stackrel{d}{=} v_{\Pi(n+1)}^{(w)*} \sim \sum_{i=1}^{n+1} p_i(X_{n+1}) \delta_{v_i^*}. \tag{16}$$

Note that $p_i$ involves the true likelihood ratio function $\omega(x)$ and thus is different from $\widehat{p}_i$. Let $\tilde{\mathbb{P}}_X$ be a measure with

$$d\tilde{\mathbb{P}}_X(x) = \widehat{\omega}(x) d\mathbb{P}_{X|W=w,e=1}(x).$$

Since $\mathbb{E}[\widehat{\omega}(x)] = 1$, $\mathbb{P}_{X \sim \mathbb{P}_{X|W=w,e=1}}(\widehat{\omega} < \infty) = 1$. As a result, $\tilde{\mathbb{P}}_X$ is a probability measure. Consider now a new sample $(\tilde{X}_{n+1}, \tilde{T}_{n+1}(w)) \sim \tilde{\mathbb{P}}_X \times \mathbb{P}_{T(w)|X,e=1}$. Let $\tilde{V}_{n+1}^{(w)}$ denote the non-conformity score of $(\tilde{X}_{n+1}, \tilde{T}_{n+1}(w))$ and set $\tilde{V}^{(w)} = (\tilde{V}_1^{(w)}, \cdots, \tilde{V}_n^{(w)}, \tilde{V}_{n+1}^{(w)})$. Using the same argument as for (16), we have

$$(\tilde{V}_{n+1}^{(w)}|\mathcal{E}(\tilde{V}^{(w)}) = \mathcal{E}(v^{(w)*}), \mathcal{D}_{\text{tr}}) \sim \sum_{i=1}^{n+1} \widehat{p}_i(X_{n+1}) \delta_{v_i^*}. \tag{17}$$

Note that each $\widehat{p}_i(\tilde{X}_{n+1})$, $i = 1, \cdots, n+1$ is well-defined since $\widehat{\omega}(X_i)$ is almost surely finite under both $\mathbb{P}_{X|W=w,e=1}$ and $\mathbb{P}_X$. As a consequence,

$$\mathbb{P}\left(\tilde{T}_{n+1}(w) \geq \widehat{L}^{(w)}(\tilde{X}_{n+1})|\mathcal{D}_{\text{tr}}\right)$$

$$= \mathbb{P}\left(\tilde{V}_{n+1}^{(w)} \leq c_{1-\alpha}^{(w)}(\tilde{X}_{n+1})|\mathcal{D}_{\text{tr}}\right)$$

$$= \mathbb{P}\left(\tilde{V}_{n+1}^{(w)} \leq \text{Quantile}\left(1-\alpha; \sum_{i=1}^n \widehat{p}(\tilde{X}_{n+1})\delta_{V_i} + \widehat{p}_\infty(\tilde{X}_{n+1})\delta_\infty\right)|\mathcal{D}_{\text{tr}}\right)$$

$$\stackrel{(i)}{=} \mathbb{P}\left(\tilde{V}_{n+1}^{(w)} \leq \text{Quantile}\left(1-\alpha; \sum_{i=1}^n \widehat{p}(\tilde{X}_{n+1})\delta_{V_i} + \widehat{p}_\infty(\tilde{X}_{n+1})\delta_{\tilde{V}_{n+1}}\right)|\mathcal{D}_{\text{tr}}\right)$$

$$= \mathbb{E}\mathbb{P}\left(\tilde{V}_{n+1}^{(w)} \leq \text{Quantile}\left(1-\alpha; \sum_{i=1}^n \widehat{p}(\tilde{X}_{n+1})\delta_{V_i} + \widehat{p}_\infty(\tilde{X}_{n+1})\delta_{\tilde{V}_{n+1}}\right)|\mathcal{E}(\tilde{V}^{(w)}), \mathcal{D}_{\text{tr}}\right)$$

$$\stackrel{(ii)}{\geq} 1 - \alpha, \tag{18}$$

where (i) uses Lemma A.2 and $(ii)$ uses (17) and the definition of $\mathrm{Quantile}(\beta; F)$. By Lemma (A.3),

$$d_{TV}\left(\mathbb{P}_X \times \mathbb{P}_{T(w)|X,e=1}, \tilde{\mathbb{P}}_X \times \mathbb{P}_{T(w)|X,e=1}\right) = d_{TV}\left(\mathbb{P}_X, \tilde{\mathbb{P}}_X\right),$$

and as a consequence,

$$\left|\mathbb{P}\left(T_{n+1}(w) \geq \widehat{L}^{(w)}(X_{n+1})|\mathcal{D}_{\mathrm{tr}}, \mathcal{D}_{\mathrm{cal}}\right) - \mathbb{P}\left(\tilde{T}_{n+1}(w) \geq \widehat{L}^{(w)}(X_{n+1})|\mathcal{D}_{\mathrm{tr}}, \mathcal{D}_{\mathrm{cal}}\right)\right| d_{TV}\left(\mathbb{P}_X, \tilde{\mathbb{P}}_X\right),$$

(19)

which implies that

$$\mathbb{P}\left(T_{n+1}(w) \geq \widehat{L}^{(w)}(X_{n+1})|\mathcal{D}_{\mathrm{tr}}, \mathcal{D}_{\mathrm{cal}}\right) \geq \mathbb{P}\left(\tilde{T}_{n+1}(w) \geq \widehat{L}^{(w)}(X_{n+1})|\mathcal{D}_{\mathrm{tr}}, \mathcal{D}_{\mathrm{cal}}\right) - d_{TV}\left(\mathbb{P}_X, \tilde{\mathbb{P}}_X\right).$$

Taking expectation over $\mathcal{D}_{\mathrm{cal}}$, we have

$$\mathbb{P}\left(T_{n+1}(w) \geq \widehat{L}^{(w)}(X_{n+1})|\mathcal{D}_{\mathrm{tr}}, \mathcal{D}_{\mathrm{cal}}\right) \geq \mathbb{P}\left(\tilde{T}_{n+1}(w) \geq \widehat{L}^{(w)}(X_{n+1})|\mathcal{D}_{\mathrm{tr}}, \mathcal{D}_{\mathrm{cal}}\right) - d_{TV}\left(\mathbb{P}_X, \tilde{\mathbb{P}}_X\right)$$
$$\geq 1 - \alpha - d_{TV}\left(\mathbb{P}_X, \tilde{\mathbb{P}}_X\right).$$

Using the integral definition of total-variation distance and (13),

$$d_{TV}\left(\mathbb{P}_X, \tilde{\mathbb{P}}_X\right) = \frac{1}{2} \int |\widehat{\omega}(x) d\mathbb{P}_{X|W=w,e=1}(x) - d\mathbb{P}_X(x)|$$
$$= \frac{1}{2} \int |\widehat{\omega}(x) d\mathbb{P}_{X|W=w,e=1}(x) - \omega(x) d\mathbb{P}_{X|W=w,e=1}(x)|$$
$$= \frac{1}{2} \mathbb{E}_{X \sim \mathbb{P}_{X|W=w,e=1}}[|\widehat{\omega}(X) - \omega(X)|].$$

Then, taking expectation over $\mathcal{D}_{\mathrm{tr}}$, we have

$$\mathbb{P}\left(T_{n+1}(w) \geq \widehat{L}^{(w)}(X_{n+1})\right) \geq 1 - \alpha - \frac{1}{2}\mathbb{E}_{X \sim \mathbb{P}_{X|W=w,e=1}}[|\widehat{\omega}(X) - \omega(X)|],$$

when $\mathbb{P}_{X \sim \mathbb{P}_X}(w(X) < \infty) = 1$.

**Step II.** Under $\mathbb{P}_{X \sim \mathbb{P}_X}(w(X) < \infty) < 1$. If $\mathbb{P}_{X \sim \mathbb{P}_{X|W=w,e=1}}(w(X) < \infty) < 1$, it is clear that $\mathbb{E}_{X \sim \mathbb{P}_{X|W=w,e=1}}[\widehat{\omega}(X) - \omega(X)] = \infty$ and Theorem 4.1 holds trivially. Thus, we assume $\mathbb{P}_{X \sim \mathbb{P}_{X|W=w,e=1}}(w(X) < \infty) = 1$ in the remainder.

Let $\mathbb{P}'_X$ denote the distribution $\mathbb{P}_X$ conditional on the event $E_\infty \triangleq \{x : \omega(x) < \infty\}$; that is,

$$d\mathbb{P}'_X(x) = \frac{I(x \in E_\infty) d\mathbb{P}_X(x)}{\mathbb{P}_{X \sim \mathbb{P}_X}(E_\infty)}$$

(20)

Further, set $\omega'(x) = d\mathbb{P}'_X(x)/d\mathbb{P}_{X|W=w,e=1}(x)$ and $\omega'(x) = \widehat{\omega}(x)I(x \in E_\infty)/\mathbb{P}_{X \sim \mathbb{P}_X}(E_\infty)$. Note that $\widehat{L}^{(w)}$ remains the same on $E_\infty$ when $\widehat{\omega}$ is replaced by $\widehat{\omega}'$ and $\mathbb{P}_X$ is replaced by $\mathbb{P}'_X$, because the weighted-split-CQR algorithm is invariant with respect to rescaling of the covariate shift estimate. Since $\mathbb{P}_{X \sim \mathbb{P}'_X}(\omega(X) < \infty) = 1$, (4) implies that

$$\mathbb{P}_{(X,T(w)) \sim \mathbb{P}'_X \times \mathbb{P}_{T(w)|X,e=1}}\left(T(w) \geq \widehat{L}^{(w)}(X_{n+1})\right) \geq 1 - \alpha - \frac{1}{2}\mathbb{E}_{X \sim \mathbb{P}_{X|W=w,e=1}}[|\widehat{\omega}'(X) - \omega'(X)|].$$

It can be reformulated as

$$\mathbb{P}\left(T_{n+1}(w) \geq \widehat{L}^{(w)}(X_{n+1})\right) \geq 1 - \alpha - \frac{1}{2\mathbb{P}_{X \sim \mathbb{P}_X}(E_\infty)}\mathbb{E}_{X \sim \mathbb{P}_{X|W=w,e=1}}[|\widehat{\omega}(X) - \omega(X)|].$$

On the other hand, when $\omega(X_{n+1}) = \infty$, $c_{1-\alpha}^{(w)}(X_{n+1}) = \infty$, implying that $\widehat{L}^{(w)}(X_{n+1}) = -\infty$. As a result,

$$\mathbb{P}\left(T_{n+1}(w) \geq \widehat{L}^{(w)}(X_{n+1})|\omega(X_{n+1}) = \infty\right) = 1.$$

Putting the two pieces together, we have that

$$\mathbb{P}\left(T_{n+1}(w) \geq \widehat{L}^{(w)}(X_{n+1})\right) = \mathbb{P}\left(T_{n+1}(w) \geq \widehat{L}^{(w)}(X_{n+1})|\omega(X_{n+1}) < \infty\right)\mathbb{P}(\omega(X_{n+1}) < \infty)$$
$$\mathbb{P}\left(T_{n+1}(w) \geq \widehat{L}^{(w)}(X_{n+1})|\omega(X_{n+1}) = \infty\right)\mathbb{P}(\omega(X_{n+1}) = \infty)$$
$$\geq (1 - \alpha)\mathbb{P}_{X \sim \mathbb{P}_X}(E_\infty) + \mathbb{P}_{X \sim \mathbb{P}_X}(E_\infty^c)$$
$$- \frac{1}{2}\mathbb{E}_{X \sim \mathbb{P}_{X|W=w,e=1}}[|\widehat{\omega}(X) - \omega(X)|]$$
$$\geq 1 - \alpha - \frac{1}{2}\mathbb{E}_{X \sim \mathbb{P}_{X|W=w,e=1}}[|\widehat{\omega}(X) - \omega(X)|].$$

$\square$

# B    PROOF OF DOUBLY ROBUSTNESS

We first prove the nonasymptotic result for one side of the double robustness and then present a simpler asymptotic result as a corollary in Appendix B.2. And then we prove Theorem 4.2 in Appendix B.3.

**Theorem B.1.** *In the setting of Theorem 4.1, further assume that*
*(1) there exists $r, b_1, b_2 > 0$, such that $\mathbb{P}(T(w) = t | X = x) \in [b_1, b_2]$ uniformly over all $(x, t)$ with $t \in [q_\alpha^{(w)}(x) - r, q_\alpha^{(w)}(x) + r]$;*
*(2) $\mathbb{P}_{X \sim \mathbb{P}_X}(\omega(X) < \infty) = 1$, and there exist $\delta, M > 0$ such that $(\mathbb{E}[\widehat{\omega}(X)^{1+\delta}])^{1/(1+\delta)} \leq M$;*
*(3) there exists $k, \ell > 0$ such that $\lim_{N \to \infty} \mathbb{E}[\widehat{\omega}(X)\mathcal{E}_N^k(X)] = \lim_{N \to \infty} \mathbb{E}[\omega(X)\mathcal{E}_N^\ell(X)] = 0$, where*

$$\mathcal{E}_N(X) = |\widehat{q}_{\beta,N}^{(w)}(x) - q_\beta^{(w)}(x)|.$$

*Then there is a constant $B_1$ that only depends on $r, b_1, b_2, \delta, M, k, \ell$ such that*

$$
\mathbb{P}_{(X, T(w)) \sim \mathbb{P}_X \times \mathbb{P}_{T(w)|X, e=1}}(T(w) \geq \widehat{L}_{N,n}^{(w)}(X))
$$
$$
\geq 1 - \alpha - B_1 \left\{ \frac{(\log n)^{(1+\delta')/2(2+\delta')}}{n^{\delta'/(2+\delta')}} + \left( \mathbb{E}[\widehat{\omega}_N(X)\mathcal{E}_N^k(X)] \right)^{(1/(2+k))} + \left( \mathbb{E}[\widehat{\omega}_N(X)\mathcal{E}_N^\ell(X)] \right)^{(1/(2+\ell))} \right\},
$$
$$(21)$$

*where $\delta' = \min\{\delta, 1\}$.*
*Furthermore, for any $\beta \in (0, 1)$ there is a constant $B_2$ that only depends on $r, b_1, b_2, \delta, M, k, \ell$ such that, with probability at least $1 - \beta$,*

$$
\mathbb{P}_{(X, T(w)) \sim \mathbb{P}_X \times \mathbb{P}_{T(w)|X, e=1}}(T(w) \geq \widehat{L}_{N,n}^{(w)}(X))
$$
$$
\geq 1 - \alpha - B_2 \left\{ \frac{(\log n)^{(1+\delta')/2(2+\delta')}}{n^{\delta'/(2+\delta')}} + \left( \mathbb{E}[\widehat{\omega}_N(X)\mathcal{E}_N^k(X)] \right)^{(1/(2+k))} + \left( \mathbb{E}[\widehat{\omega}_N(X)\mathcal{E}_N^\ell(X)] \right)^{(1/(2+\ell))} \right\}.
$$
$$(22)$$

## B.1    PROOF OF THEOREM B.1

We start with the following two Rosenthal-type inequalities for sums of independent random variables with finite $(1 + \delta)$-th moments.

**Proposition B.2.** *(Theorem 3 of Rosenthal (1970)). Let $\{Z_i\}_{i:1,..,n}$ be independent mean-zero random variables. Then for any $\delta \geq 1$, there exists $L(\delta) > 0$ that only depends on $\delta$ such that*

$$
\mathbb{E}\left| \sum_{i=1}^n Z_i \right|^{1+\delta} \leq L(\delta)\left\{ \sum_{i=1}^n \mathbb{E}|Z_i|^{1+\delta} + \left( \sum_{i=1}^n \mathbb{E}|Z_i|^2 \right)^{(1+\delta)/2} \right\}.
$$

**Proposition B.3.** *(Theorem 2 of von Bahr & Esseen (1965)). Let $\{Z_i\}_{i:1,..,n}$ be independent mean-zero random variables. Then for any $\delta \in [0, 1)$,*

$$
\mathbb{E}\left| \sum_{i=1}^n Z_i \right|^{1+\delta} \leq 1 \sum_{i=1}^n \mathbb{E}|Z_i|^{1+\delta}.
$$

*Proof.* For notational convenience, we suppress the subscripts $N$ and $n$ in $\widehat{q}^{(w)}, \widehat{\omega}, \widehat{L}^{(w)}$ as well as in $\widehat{p}_i(x)$ and $c_{1-\alpha}^{(w)}(x)$. Note that Assumption (2) implies that $\omega(X)$ is almost surely finite under $\mathbb{P}_X$ and $\widehat{\omega}(X)$ is almost surely finite under $\mathbb{P}_{X|W=w, e=1}$. By the same reasoning as in the Section A.1, $\omega(X)$ is almost surely finite under $\mathbb{P}_{X|W=w, e=1}$ and $\widehat{\omega}(X)$ is almost surely finite under $\mathbb{P}_X$.
Let $\epsilon < r/2$ and $(\tilde{X}, \tilde{T}(w))$ denote a generic random vector drawn from $\mathbb{P}_X \times \mathbb{P}_{T(w)|X, e=1}$, which is

independent of the data.Then

$$
\begin{aligned}
&\mathbb{P}(\tilde{T}(w) \geq \widehat{L}(\tilde{X})|\tilde{X}) \\
&= \mathbb{P}\left(\widehat{q}_\alpha^{(w)}(\tilde{X}) - \tilde{T}(w) \leq c_{1-\alpha}^{(w)}(\tilde{X})|\tilde{X}\right) \\
&\geq \mathbb{P}\left(q_\alpha^{(w)}(\tilde{X}) - \tilde{T}(w) \leq c_{1-\alpha}^{(w)}(\tilde{X}) - \mathcal{E}(\tilde{X})|\tilde{X}\right) \\
&\geq \mathbb{P}\left(q_\alpha^{(w)}(\tilde{X}) - \tilde{T}(w) \leq -\epsilon - \mathcal{E}(\tilde{X})|\tilde{X}\right) - \mathbb{P}(c_{1-\alpha}^{(w)}(\tilde{X}) < -\epsilon|\tilde{X}) \\
&\geq \mathbb{P}\left(q_\alpha^{(w)}(\tilde{X}) - \tilde{T}(w) \leq -\epsilon - \mathcal{E}(\tilde{X})I(\mathcal{E}(\tilde{X}) \leq \epsilon)|\tilde{X}\right) - I(\mathcal{E}(\tilde{X}) > \epsilon) - \mathbb{P}(c_{1-\alpha}^{(w)}(\tilde{X}) < -\epsilon|\tilde{X}) \\
&\overset{(i)}{\geq} \mathbb{P}\left(q_\alpha^{(w)}(\tilde{X}) - \tilde{T}(w) \leq 0\right) - b_2\left\{\epsilon + \mathcal{E}(\tilde{X})I(\mathcal{E}(\tilde{X}) \leq \epsilon)\right\} - I(\mathcal{E}(\tilde{X}) > \epsilon) - \mathbb{P}(c_{1-\alpha}^{(w)}(\tilde{X}) < -\epsilon|\tilde{X}) \\
&\geq \mathbb{P}\left(q_\alpha^{(w)}(\tilde{X}) - \tilde{T}(w) \leq 0\right) - b_2\left\{\epsilon + \mathcal{E}(\tilde{X}))\right\} - I(\mathcal{E}(\tilde{X})) > \epsilon) - \mathbb{P}(c_{1-\alpha}^{(w)}(\tilde{X}) < -\epsilon|\tilde{X}) \\
&\overset{(ii)}{=} 1 - \alpha - b_2\left\{\epsilon + \mathcal{E}(\tilde{X}))\right\} - I(\mathcal{E}(\tilde{X})) > \epsilon) - \mathbb{P}(c_{1-\alpha}^{(w)}(\tilde{X}) < -\epsilon|\tilde{X});
\end{aligned}
\tag{23}
$$

above, (i) uses the condition that $\epsilon < r/2$, Assumption (1) and the definitions of $q_\alpha^{(w)}$ that $\mathbb{P}(\tilde{T}(w) \geq q_\alpha^{(w)}(\tilde{X})) = 1 - \alpha$.

Next, we derive an upper bound on $\mathbb{P}(c_{1-\alpha}^{(w)}(\tilde{X}) < -\epsilon|\tilde{X})$. Let $G$ denote the cumulative distribution function of the random distribution $\sum_{i=1}^n \widehat{p}_i(\tilde{X})\delta_{V_i} + \widehat{p}_\infty(\tilde{X})\delta_\infty$. Again, $G$ implicitly depends on $N$, $n$, and $\tilde{X}$. Then $c_{1-\alpha}^{(w)} < -\epsilon$ implies $G(-\epsilon) \geq 1 - \alpha$, and thus,

$$
\mathbb{P}\left(c_{1-\alpha}^{(w)}(\tilde{X}) < -\epsilon|\tilde{X}\right) \leq \mathbb{P}\left(G(-\epsilon) \geq 1 - \alpha|\tilde{X}\right), \text{a.s..}
$$

Let $G^*(-\epsilon)$ denote the expectation of $G(-\epsilon)$ conditional on $\mathcal{D} = \{\mathcal{D}_{\text{tr}}, (X_i)_{i=1}^n, \tilde{X}\}$, namely,

$$
G^*(-\epsilon) = \mathbb{E}[G(-\epsilon)|\mathcal{D}] = \sum_{i=1}^n \widehat{p}_i(\tilde{X})\mathbb{P}(V_i^{(w)} \leq -\epsilon|\mathcal{D}).
$$

For any $s > 0$, the triangle inequality implies that

$$
\mathbb{P}(c_{1-\alpha}^{(w)}(\tilde{X}) < -\epsilon) \leq \mathbb{P}(G(-\epsilon) - G^*(-\epsilon) \geq s|\tilde{X}) + \mathbb{P}(G^*(-\epsilon) \geq 1 - s|\tilde{X}), \text{a.s..}
\tag{24}
$$

To bound the first term, we note that

$$
G(-\epsilon) - G^*(-\epsilon) = \sum_{i=1}^n \widehat{p}_i(\tilde{X})(I(V_i^{(w)} \leq -\epsilon) - \mathbb{P}(V_i \leq -\epsilon|\mathcal{D})).
$$

Conditional on $\mathcal{D}$, $G(-\epsilon) - G^*(-\epsilon)$ is sub-Gaussian with parameter

$$
\widehat{\sigma}^2 = \sum_{1+n}^n \widehat{p}_i(\tilde{X})^2.
$$

For any $s > 0$,

$$
\mathbb{P}(G(-\epsilon) - G^*(-\epsilon) \geq s|\mathcal{D}) \leq \exp\left(-\frac{s^2}{2\widehat{\sigma}^2}\right).
$$

Let $\gamma_n$ be any fixed sequence with $\gamma_n = O(1)$. Taking expectation over $\mathcal{D}\backslash\{\tilde{X}\}$, we obtain that

$$\mathbb{P}\left(G(-\epsilon) - G^*(-\epsilon) \geq s|\tilde{X}\right)$$

$$\leq \mathbb{E}\left[\exp\left(-\frac{s^2}{2\widehat{\sigma}^2}\right)|\tilde{X}\right]$$

$$\leq \exp\left(-\frac{s^2}{2\widehat{\sigma}^2}\right) + \mathbb{P}(\widehat{\sigma}^2 \geq \gamma_n|\tilde{X})$$

$$= \exp\left(-\frac{s^2}{2\widehat{\sigma}^2}\right) + \mathbb{P}\left(\frac{\sum_{i=1}^n \widehat{\omega}(\tilde{X}_i)^2}{\left(\sum_{i=1}^n \widehat{\omega}(\tilde{X}_i) + \widehat{\omega}(\tilde{X})\right)^2} \geq \gamma_n|\tilde{X}\right)$$

$$\leq \exp\left(-\frac{s^2}{2\gamma_n}\right) + \mathbb{P}\left(\frac{\sum_{i=1}^n \widehat{\omega}(X_i)^2}{\left(\sum_{i=1}^n \widehat{\omega}(\tilde{X}_i)\right)^2} \geq \gamma_n|\tilde{X}\right)$$

$$\overset{(i)}{=} \exp\left(-\frac{s^2}{2\gamma_n}\right) + \mathbb{P}\left(\frac{\sum_{i=1}^n \widehat{\omega}(X_i)^2}{\left(\sum_{i=1}^n \widehat{\omega}(\tilde{X}_i)\right)^2} \geq \gamma_n\right)$$

$$\leq \exp\left(-\frac{s^2}{2\gamma_n}\right) + \mathbb{P}\left(\sum_{i=1}^n \widehat{\omega}(X_i) \leq \frac{n}{2}\right) + \mathbb{P}\left(\sum_{i=1}^n \widehat{\omega}(X_i)^2 \geq \frac{n^2\gamma_n}{4}\right)$$

$$\leq \exp\left(-\frac{s^2}{2\gamma_n}\right) + \mathbb{P}\left(\sum_{i=1}^n |\widehat{\omega}(X_i) - 1| \geq \frac{n}{2}\right) + \mathbb{P}\left(\sum_{i=1}^n \widehat{\omega}(X_i)^2 \geq \frac{n^2\gamma_n}{4}\right),$$

where (i) uses the fact that $\tilde{X}$ is independent of $(\widehat{\omega}(X_i))_{i=1}^n$. Note that this bound holds uniformly with $\tilde{X}$. Throughout the rest of the proof, we write $a_{1n} \lesssim a_{2n}$ if there exists a constant $B$ that only depends on $r, b_1, b_2, \delta, M, k, \ell$ such that $a_{1n} \leq Ba_{2n}$ for all $n$. We consider two cases:

(w) If $\delta \geq 1$, then by Markov's inequality,

$$\mathbb{P}\left(\sum_{i=1}^n \widehat{\omega}(X_i)^2 \geq \frac{n^2\gamma_n}{4}\right) \leq \frac{4\mathbb{E}[\sum_{i=1}^n \widehat{\omega}(X_i)^2]}{n^2\gamma_n} = \frac{4\mathbb{E}[\widehat{\omega}(X_1)^2]}{n\gamma_n} \lesssim \frac{1}{n\gamma_n}.$$

Since $\mathbb{E}[\widehat{\omega}(X_i)|\mathcal{D}_{\text{tr}}] = 1$, we have $\mathbb{E}[\widehat{\omega}(X_i)] = 1$. By Markov's inequality and Proposition B.3,

$$\mathbb{P}\left(\sum_{i=1}^n |\widehat{\omega}(X_i) - 1| \geq \frac{n}{2}\right)$$

$$\geq \frac{2^{1+\delta}}{n^{1+\delta}}\mathbb{E}\left(\sum_{i=1}^n |\widehat{\omega}(X_i) - \mathbb{E}[\widehat{\omega}(X_i)]|\right)^{1+\delta}$$

$$\lesssim \frac{1}{n^{1+\delta}}\left\{n\mathbb{E}|\widehat{\omega}(X_i) - \mathbb{E}[\widehat{\omega}(X_i)]|^{1+\delta} + n^{(1+\delta)/2}(\mathbb{E}|\widehat{\omega}(X_i) - \mathbb{E}[\widehat{\omega}(X_i)]|^2)^{(1+\delta)/2}\right\}$$

$$\overset{(i)}{\lesssim} \frac{1}{n^{1+\delta}}\left\{n\mathbb{E}|\widehat{\omega}(X_i)|^{1+\delta} + n^{(1+\delta)/2}(\mathbb{E}|\widehat{\omega}(X_i)|^2)^{(1+\delta)/2}\right\}$$

$$\lesssim \frac{1}{n^{(1+\delta)/2}} \qquad (25)$$

where (i) follows from Hölder's inequality which gives

$$\mathbb{E}|\widehat{\omega}(X_i) - \mathbb{E}[\widehat{\omega}(X_i)]|^{1+\delta} \leq 2^\delta\left(\mathbb{E}|\widehat{\omega}(X_i)|^{1+\delta} + \mathbb{E}[\widehat{\omega}(X_i)]|^{1+\delta}\right) \leq 2^{1+\delta}\mathbb{E}|\widehat{\omega}(X_i)|^{1+\delta}.$$

Piecing things together yields

$$\mathbb{P}\left(G(-\epsilon) - G^*(-\epsilon) \geq s\right) \lesssim \exp\left(-\frac{s^2}{2\gamma_n}\right) + \frac{1}{n^{(1+\delta)/2}} + \frac{1}{n\gamma_n} \lesssim \exp\left(-\frac{s^2}{2\gamma_n}\right) + \frac{1}{n\gamma_n},$$

where the last step follows from the fact that $\delta \geq 1$ and $\gamma_n = O(1)$.

(2) If $\delta < 1$, then by Markov's inequality,

$$\mathbb{P}\left(\sum_{i=1}^n \widehat{\omega}(X_i)^2 \geq \frac{n^2\gamma_n}{4}\right) \leq \frac{\mathbb{E}\left[(\sum_{i=1}^n \widehat{\omega}(X_i)^2)^{(1+\delta)/2}\right]}{(n^2\gamma_n)^{(1+\delta)/2}} \leq \frac{\mathbb{E}\left[(\sum_{i=1}^n \widehat{\omega}(X_i)^2)^{(1+\delta)/2}\right]}{(n^2\gamma_n)^{(1+\delta)/2}},$$

where the last step follows from the simple fact that $\|x\|_p \leq \|x\|_1$ for $p \geq 1$, with $p = 1/(1+\delta)$ and $x_i = \widehat{\omega}(X_i)^{(1+\delta)}$. By Markov's inequality and Proposition B.3,

$$\mathbb{P}\left(\sum_{i=1}^n |\widehat{\omega}(X_i) - 1| \geq \frac{n}{2}\right) \leq \frac{2^{1+\delta}}{n^{1+\delta}}\mathbb{E}\left(\sum_{i=1}^n |\widehat{\omega}(X_i) - \mathbb{E}[\widehat{\omega}(X_i)]|\right)^{1+\delta}$$

$$\leq \frac{2n\mathbb{E}|\widehat{\omega}(X_i) - \mathbb{E}[\widehat{\omega}(X_i)]|^{1+\delta}}{n^{1+\delta}} \leq \frac{1}{n^\delta}. \tag{26}$$

Piecing things together yields

$$\mathbb{P}\left(G(-\epsilon) - G^*(-\epsilon) \geq s|\tilde{X}\right) \leq \exp\left(-\frac{s^2}{2\gamma_n}\right) + \frac{1}{n^\delta} + \frac{1}{n^\delta \gamma_n^{(1+\delta)/2}}$$

$$\leq \exp\left(-\frac{s^2}{2\gamma_n}\right) + \frac{1}{n^\delta \gamma_n^{(1+\delta)/2}},$$

where the last step follows from $\gamma_n = O(1)$.

In all cases,

$$\mathbb{P}\left(G(-\epsilon) - G^*(-\epsilon) \geq s|\tilde{X}\right) \leq \exp\left(-\frac{s^2}{2\gamma_n}\right) + \frac{1}{n^{\delta'} \gamma_n^{(1+\delta')/2}}, \quad \delta' = \min\{\delta, 1\}. \tag{27}$$

Next, we almost surely bound the term $\mathbb{P}\left(G^*(-\epsilon) \geq 1 - s|\tilde{X}\right)$. By the triangle inequality and definition of $\mathcal{E}(\cdot)$,

$$V_i^{(w)} \geq \widehat{q}_\alpha^{(w)}(X_i) - T_i(w) - \mathcal{E}(X_i) \triangleq V_i^{(w)*} - \mathcal{E}(X_i).$$

By Assumption (1) and the definitions of $q_\alpha^{(w)}$ that $\mathbb{P}(\tilde{T} \geq q_\alpha^{(w)}(\tilde{X})) = 1-\alpha$, $\mathbb{P}(V_i^{(w)*} \leq 0|\mathcal{D}) = 1-\alpha$. Conditional on $\mathcal{D}$, $\mathcal{E}(X_i)$ is deterministic. Since $\epsilon < r/2 < 2r$,

$$G^*(-\epsilon) \leq \sum_{i=1}^n \widehat{p}_i(\tilde{X})\left\{I\left(\mathcal{E}(X_i) \geq \frac{\epsilon}{2}\right) + \mathbb{P}\left(V_i^{(w)*} \leq \frac{\epsilon}{2}|\mathcal{D}\right)\right\}$$

$$\leq \sum_{i=1}^n \widehat{p}_i(\tilde{X})\left\{I\left(\mathcal{E}(X_i) \geq \frac{\epsilon}{2}\right) + \mathbb{P}\left(V_i^{(w)*} \leq 0|\mathcal{D}\right) - \frac{\epsilon b_1}{2}\right\}$$

$$= \sum_{i=1}^n \widehat{p}_i(\tilde{X})\left\{I\left(\mathcal{E}(X_i) \geq \frac{\epsilon}{2}\right) + 1 - \alpha - \frac{\epsilon b_1}{2}\right\}$$

$$\leq 1 - \alpha - \frac{\epsilon b_1}{2} + \sum_{i=1}^n \widehat{p}_i(\tilde{X})I\left(\mathcal{E}(X_i) \geq \frac{\epsilon}{2}\right), \tag{28}$$

where the last step follows from the fact that $\sum_{i=1}^{n} \widehat{p}_i(\tilde{X}) \leq 1$. If $s \leq \epsilon b_1/4$, (28) implies that

$$\mathbb{P}\left(G^*(-\epsilon) \geq 1 - \alpha - s|\tilde{X}\right)$$

$$\leq \mathbb{P}\left(\sum_{i=1}^{n} \widehat{p}_i(\tilde{X})I\left(\mathcal{E}(X_i) \geq \frac{\epsilon}{2}\right) \geq \frac{\epsilon b_1}{2} - s|\tilde{X}\right)$$

$$= \mathbb{P}\left(\frac{\sum_{i=1}^{n} \widehat{\omega}(X_i)I(\mathcal{E}(X_i) \geq \epsilon/2)}{\sum_{i=1}^{n} \widehat{\omega}(X_i) + \widehat{\omega}(\tilde{X})} \geq \frac{\epsilon b_1}{2} - s|\tilde{X}\right)$$

$$\leq \mathbb{P}\left(\frac{\sum_{i=1}^{n} \widehat{\omega}(X_i)I(\mathcal{E}(X_i) \geq \epsilon/2)}{\sum_{i=1}^{n} \widehat{\omega}(X_i)} \geq \frac{\epsilon b_1}{2} - s|\tilde{X}\right)$$

$$\overset{(i)}{=} \mathbb{P}\left(\frac{\sum_{i=1}^{n} \widehat{\omega}(X_i)I(\mathcal{E}(X_i) \geq \epsilon/2)}{\sum_{i=1}^{n} \widehat{\omega}(X_i)} \geq \frac{\epsilon b_1}{2} - s\right)$$

$$\leq \mathbb{P}\left(\sum_{i=1}^{n} \widehat{\omega}(X_i) \leq \frac{n}{2}\right) + \mathbb{P}\left(\sum_{i=1}^{n} \widehat{\omega}(X_i)I\left(\mathcal{E}(X_i) \geq \frac{\epsilon}{2}\right) \geq \frac{n(\epsilon b_1 - 2s)}{4}\right)$$

$$\overset{(ii)}{\leq} \mathbb{P}\left(\sum_{i=1}^{n} |\widehat{\omega}(X_i) - 1| \geq \frac{n}{2}\right) + \mathbb{P}\left(\sum_{i=1}^{n} \widehat{\omega}(X_i)I\left(\mathcal{E}(X_i) \geq \frac{\epsilon}{2}\right) \geq \frac{n\epsilon b_1}{8}\right),$$

where (i) follows from the independence between $\tilde{X}$ and $\{\mathcal{D} \setminus \{\tilde{X}\}\}$ and (ii) follows from the fact that $s \leq \epsilon b_1/4$. By (25) and (26), we have that

$$\mathbb{P}\left(\sum_{i=1}^{n} |\widehat{\omega}(X_i) - 1| \geq \frac{n}{2}\right) \leq \frac{1}{n^{(\delta+\delta')/2}},$$

where $\delta' = \min\{\delta, 1\}$. By Markov's inequality,

$$\mathbb{P}\left(\sum_{i=1}^{n} \widehat{\omega}(X_i)I\left(\mathcal{E}(X_i) \geq \frac{\epsilon}{2}\right) \geq \frac{n\epsilon b_1}{8}\right) \leq \frac{1}{\epsilon}\mathbb{E}\left[\widehat{\omega}(X)I\left(\mathcal{E}(X_i) \geq \frac{\epsilon}{2}\right)\right] \leq \frac{\mathbb{E}[\widehat{\omega}(X)\mathcal{E}^k(X)]}{\epsilon^{1+k}},$$

where the last step uses the simple fact that $I(\mathcal{E}(X_i) \geq \epsilon/2) \leq (2/\epsilon)^k \mathcal{E}^k(X_i)$. Therefore, for any $s \leq \epsilon b_1/4$, we obtain an almost sure bound of the form

$$\mathbb{P}\left(G^*(-\epsilon) \geq 1 - \alpha - s|\tilde{X}\right) \leq \frac{1}{n^{(\delta+\delta')/2}} + \frac{\mathbb{E}[\widehat{\omega}(X)\mathcal{E}^k(X)]}{\epsilon^{1+k}}. \tag{29}$$

Combining (24), (27) and (29) together and setting $s = \epsilon b_1/4$, we obtain that for any sequence $\gamma_n = O(1)$,

$$\mathbb{P}\left(c_{1-\alpha}^{(w)}(\tilde{X}) < -\epsilon\right) \leq \exp\left(-\frac{b_1^2}{32}\frac{\epsilon^2}{\gamma_n}\right) + \frac{1}{n^{\delta'}\gamma_n^{(1+\delta')/2}} + \frac{1}{n^{(\delta+\delta')/2}} + \frac{\mathbb{E}[\widehat{\omega}(X)\mathcal{E}^k(X)]}{\epsilon^{1+k}}$$

$$\leq \exp\left(-\frac{b_1^2}{32}\frac{\epsilon^2}{\gamma_n}\right) + \frac{1}{n^{\delta'}\gamma_n^{(1+\delta')/2}} + \frac{\mathbb{E}[\widehat{\omega}(X)\mathcal{E}^k(X)]}{\epsilon^{1+k}}. \tag{30}$$

Substitute $\epsilon$ with $\epsilon_n$ and assume $\epsilon_n \leq r/2$ (recall the beginning of proof). Set

$$\gamma_n = \frac{b_1^2}{32}\frac{\epsilon_n^2}{\log n}. \tag{31}$$

Clearly, $\gamma_n = o(1)$. Then the first term of (30) is $1/n$, and thus,

$$\mathbb{P}\left(c_{1-\alpha}^{(w)}(\tilde{X}) < -\epsilon_n|\tilde{X}\right) \leq \frac{(\log n)^{(1+\delta')/2}}{n^{\delta'}\epsilon_n^{1+\delta'}} + \frac{\mathbb{E}[\widehat{\omega}(X)\mathcal{E}^k(X)]}{\epsilon_n^{1+k}}.$$

Equivalently, there exists a constant $B$ that only depends on $r, b_1, b_2, \delta, M, k, \ell$, such that

$$\mathbb{P}\left(c_{1-\alpha}^{(w)}(\tilde{X}) < -\epsilon_n|\tilde{X}\right) \leq B\left\{\frac{(\log n)^{(1+\delta')/2}}{n^{\delta'}\epsilon_n^{1+\delta'}} + \frac{\mathbb{E}[\widehat{\omega}(X)\mathcal{E}^k(X)]}{\epsilon_n^{1+k}}\right\}, a.s..$$

Together with (23), it implies that

$$\mathbb{P}(\tilde{T}(w) \geq \widehat{L}^{(w)}(\tilde{X})|\tilde{X})$$

$$\geq 1 - \alpha - b_2(\epsilon_n + \mathcal{E}(\tilde{X})) - I(\mathcal{E}(\tilde{X}) > \epsilon_n) - B\left\{\frac{(\log n)^{(1+\delta')/2}}{n^{\delta'}\epsilon_n^{1+\delta'}} + \frac{\mathbb{E}[\widehat{\omega}(X)\mathcal{E}^k(X)]}{\epsilon_n^{1+k}}\right\},$$

almost surely. Assume $B \geq 2b_2$ without loss of generality. Then

$$\mathbb{P}\left(\mathbb{P}(\tilde{T}(w) \geq \widehat{L}^{(w)}(\tilde{X})|\tilde{X}) \leq 1 - \alpha - B\left\{\frac{(\log n)^{(1+\delta')/2}}{n^{\delta'}\epsilon_n^{1+\delta'}} + \frac{\mathbb{E}[\widehat{\omega}(X)\mathcal{E}^k(X)]}{\epsilon_n^{1+k}}\right\}\right) \leq \mathbb{P}(\mathcal{E}(\tilde{X}) > \epsilon_n). \tag{32}$$

For any $\beta \in (0,1)$, let

$$\epsilon_n = \frac{(\log n)^{(1+\delta')/2(2+\delta')}}{n^{\delta'/(2+\delta')}} + \left(\mathbb{E}[\widehat{\omega}(X)\mathcal{E}^k(X)]\right)^{1/(2+k)} + \frac{\left(\mathbb{E}[\omega(X)\mathcal{E}^\ell(X)]\right)^{1/l}}{\beta^{1/l}}.$$

Then

$$\frac{(\log n)^{(1+\delta')/2}}{n^{\delta'}\epsilon_n^{1+\delta'}}, \frac{\mathbb{E}[\widehat{\omega}(X)\mathcal{E}^k(X)]}{\epsilon_n^{1+k}} \leq \epsilon_n, \tag{33}$$

and by Markov's inequality and (13),

$$\mathbb{P}(\mathcal{E}(\tilde{X}) > \epsilon_n) \leq \frac{\mathbb{E}[\mathcal{E}^\ell(\tilde{X})]}{\epsilon_n^\ell} = \frac{\mathbb{E}[\omega(X)\mathcal{E}^\ell(X)]}{\epsilon_n^\ell} \leq \beta.$$

Furthermore, Assumption (3) implies that $\epsilon_n \leq r/2$ when $N \geq N(r)$ and $n \geq n(r)$ for some constants $N(r), n(r)$ that only depend on $r$. Replacing $B$ by $3B$, we obtain that, for $N \geq N(r), n \geq n(r)$,

$$\mathbb{P}\left(\mathbb{P}(\tilde{T} \geq \widehat{L}^{(w)}(\tilde{X})|\tilde{X}) \leq 1 - \alpha - B\epsilon_n\right) \leq \beta.$$

We can further enlarge $B$ so that $B\epsilon_n \geq 1 - \alpha$ when $N < N(r)$ or $n < n(r)$, in which case (22) holds trivially.

To prove the unconditional result, we note that (32) implies

$$\mathbb{P}(\tilde{T}(w) \geq \widehat{L}^{(w)}(\tilde{X}))$$

$$\geq \left(1 - \alpha - B\left\{\epsilon_n + \frac{(\log n)^{(1+\delta')/2}}{n^{\delta'}\epsilon_n^{1+\delta'}} + \frac{\mathbb{E}[\widehat{\omega}(X)\mathcal{E}^k(X)]}{\epsilon_n^{1+k}}\right\}\right)(1 - \mathbb{P}(\mathcal{E}(\tilde{X}) > \epsilon_n)).$$

Let

$$\epsilon_n = \frac{(\log n)^{(1+\delta')/2(2+\delta')}}{n^{\delta'/(2+\delta')}} + \left(\mathbb{E}[\widehat{\omega}(X)\mathcal{E}^k(X)]\right)^{1/(2+k)} + \left(\mathbb{E}[\omega(X)\mathcal{E}^\ell(X)]\right)^{1/(1+\ell)}.$$

Then (33) remains to hold. By Markov's inequality and (13),

$$\mathbb{P}(\mathcal{E}(\tilde{X}) > \epsilon_n) \leq \frac{\mathbb{E}[\mathcal{E}^\ell(\tilde{X})]}{\epsilon_n^\ell} = \frac{\mathbb{E}[\omega(X)\mathcal{E}^\ell(X)]}{\epsilon_n^\ell} \leq \epsilon_n.$$

Furthermore, Assumption (3) implies that $\epsilon_n \leq r/2$ when $N$ and $n$ are sufficiently large, in which case,

$$\mathbb{P}(\tilde{T}(w) \geq \widehat{L}^{(w)}(\tilde{X})) \geq (1 - \alpha - 3B\epsilon_n)(1 - \epsilon_n) \geq 1 - \alpha - (3B+1)\epsilon_n. \tag{34}$$

Similar to (22), we can enlarge the constant to make (21) hold when $N$ or $n$ is not sufficiently large. □

## B.2 ASYMPTOTIC RESULT

Theorem 4.1 and Theorem B.1 together imply the following asymptotic result, which is a generalization of Theorem 4.2 in Section 4.2.

**Corollary B.4.** *With the same notation as in Theorem 4.1, assume that either **B1** or **B2** (or both) is satisfied:*
**B1** $\lim_{N\to\infty} \mathbb{E}[\widehat{\omega}_N(X) - \omega(X)] = 0$;

**B2** *the assumptions (1)-(3) and the definition of $\widehat{q}_\alpha^{(w)}$ in Theorem B.1 hold.*
*Then*

$$\lim_{N,n\to\infty} \mathbb{P}_{(X,T(w))\sim\mathbb{P}_X\times\mathbb{P}_{T(w)|X,e=1}}(T(w) \geq \widehat{L}_{N,n}^{(w)}(X)) \geq 1 - \alpha \tag{35}$$

*Furthermore, under **B2**, for any $\epsilon > 0$,*

$$\lim_{N,n\to\infty} \mathbb{P}_{X\sim\mathbb{P}_X}\left(\mathbb{P}(T(w) \geq \widehat{L}_{N,n}^{(w)}(X)|X) \leq 1 - \alpha - \epsilon\right) = 0. \tag{36}$$

## B.3 PROOF OF THEOREM 4.2

*Proof.* Since $\mathbb{E}[1/\gamma_n(X)|\mathcal{D}_{\mathrm{tr}}] < \infty$ and $\mathbb{E}[1/\gamma(X)] < \infty$, we can here set

$$\widehat{\omega}_N(x) = \frac{p(W = w, e = 1)/\widehat{\gamma}_N(x)}{\mathbb{E}[p(W = w, e = 1)/\widehat{\gamma}_N(X)|\mathcal{D}_{\mathrm{tr}}]} = \frac{1/\widehat{\gamma}_N(x)}{\mathbb{E}[1/\widehat{\gamma}_N(X)|\mathcal{D}_{\mathrm{tr}}]},$$

$$\omega(x) = \frac{d\mathbb{P}_X(x)}{d\mathbb{P}_{X|W=w,e=1}(x)} = \frac{p(W = w, e = 1)/\gamma(x)}{\mathbb{E}[p(W = w, e = 1)/\gamma(X)]} = \frac{1/\gamma(x)}{\mathbb{E}[1/\gamma(X)]},$$

$$\implies \mathbb{E}[\widehat{\omega}_N(X)|\mathcal{D}_{\mathrm{tr}}] = 1 = \mathbb{E}[\omega(X)].$$

Thus, the assumption **B1** of Corollary B.4 reduce to

$$\lim_{N\to\infty} \mathbb{E}\left|\frac{1/\widehat{\gamma}_N(x)}{\mathbb{E}[1/\widehat{\gamma}_N(X)|\mathcal{D}_{\mathrm{tr}}]} - \frac{1/\gamma(x)}{\mathbb{E}[1/\gamma(X)]}\right| = 0,$$

and the assumptions (2) and (3) in **B2** of Corollary B.4 reduce to

$$\limsup_{N\to\infty} \frac{\mathbb{E}[1/\widehat{\gamma}(X)^{1+\delta}]}{(\mathbb{E}[1/\widehat{\gamma}(X)])^{1+\delta}} < \infty, \lim_{N\to\infty} \frac{\mathbb{E}[\mathcal{E}_N(X)/\widehat{\gamma}_N(X)]}{\mathbb{E}[1/\widehat{\gamma}_N(X)]} = \lim_{N\to\infty} \frac{\mathbb{E}[\mathcal{E}_N(X)/\gamma(X)]}{\mathbb{E}[1/\gamma(X)]} = 0.$$

Clearly, **A2** implies **B2** since $e(x), \widehat{\gamma}_N(x) \in [0, 1]$. Now we prove that **A1** implies **B1**. In fact,

$$\lim_{N\to\infty} \mathbb{E}\left|\frac{1/\widehat{\gamma}_N(x)}{\mathbb{E}[1/\widehat{\gamma}_N(X)|\mathcal{D}_{\mathrm{tr}}]} - \frac{1/\gamma(x)}{\mathbb{E}[1/\gamma(X)]}\right|$$

$$\leq \limsup_{N\to\infty} \frac{1}{\mathbb{E}[1/\widehat{\gamma}_N(X)|\mathcal{D}_{\mathrm{tr}}]}\mathbb{E}\left|\frac{1}{\widehat{\gamma}_N(X)} - \frac{1}{\gamma(X)}\right|$$

$$+ \limsup_{N\to\infty} \mathbb{E}\left[\frac{1}{\gamma(X)}\right]\mathbb{E}\left|\frac{1}{\mathbb{E}[1/\widehat{\gamma}_N(X)|\mathcal{D}_{\mathrm{tr}}]} - \frac{1}{\mathbb{E}[1/\gamma(X)]}\right|$$

$$\overset{(i)}{\leq} \limsup_{N\to\infty} \mathbb{E}\left|\frac{1}{\widehat{\gamma}_N(X)} - \frac{1}{\gamma(X)}\right| + \limsup_{N\to\infty} \mathbb{E}\left[\frac{1}{\gamma(X)}\right]\mathbb{E}\left|\mathbb{E}\left[\frac{1}{\widehat{\gamma}_N(X)}|\mathcal{D}_{\mathrm{tr}}\right] - \mathbb{E}\left[\frac{1}{\gamma(X)}\right]\right|$$

$$\leq \limsup_{N\to\infty} \mathbb{E}\left|\frac{1}{\widehat{\gamma}_N(X)} - \frac{1}{\gamma(X)}\right| + \limsup_{N\to\infty} \mathbb{E}\left[\frac{1}{\gamma(X)}\right]\mathbb{E}\left(\mathbb{E}\left[\left|\frac{1}{\widehat{\gamma}_N(X)} - \frac{1}{\gamma(X)}\right||\mathcal{D}_{\mathrm{tr}}\right]\right)$$

$$\overset{(ii)}{=} \left(1 + \mathbb{E}\left[\frac{1}{\gamma(X)}\right]\right)\limsup_{N\to\infty} \mathbb{E}\left|\frac{1}{\widehat{\gamma}_N(X)} - \frac{1}{\gamma(X)}\right|;$$

(i) above uses the fact that $e(x), \widehat{\gamma}_N(x) \in [0, 1]$ and (ii) uses Assumption **A1** and the condition that $\mathbb{E}[1/\gamma(X)] < \infty$. □

Table 2: Parameters utilized in the six settings of synthetic datasets.

| Setting | $p$ | $\mu(x)$ | $\sigma(x)$ | $C(x)$ |
|---|---|---|---|---|
| 1 | 5 | $\sqrt{x_2}$ | 1 | Exp(2) |
| 2 | 5 | $(3-1.5\sqrt{x_3})\cdot\mathbb{I}\{x_2\leq 2\}+\sqrt{x_3}\cdot\mathbb{I}\{x_2\geq 2\}$ | 2 | Exp(3) |
| 3 | 10 | $\sqrt{x_4}\cdot\mathbb{I}\{x_2\leq 2\}+(2-\sqrt{x_6})\cdot\mathbb{I}\{x_2\geq 2\}$ | 1.5 | $\mathrm{Exp}\left(\frac{1}{\frac{4-\sqrt{x_2}}{40}+0.5}\right)$ |
| 4 | 10 | $2.5\cdot\mathbb{I}\{x_2\leq 2\}+x_3\cdot\mathbb{I}\{x_2\geq 2\}$ | 1.5 | $\mathrm{lognormal}\left(3+\frac{2-x_2}{50},2\right)$ |
| 5 | 20 | $0.1\left(x_2+\min(x_3x_4,x_5x_6)\right)$ | 2 | $\mathrm{Exp}\left(\frac{1}{\frac{x_2+x_8}{50}+0.6}\right)$ |
| 6 | 50 | $0.1\left(x_2+\sqrt{x_4x_6}\right)$ | $\frac{x_3+2}{2}$ | $\mathrm{Exp}\frac{1}{\frac{x_{20}+6}{20}}$ |

Table 3: The censored and treated rate of synthetic datasets, with $W\in\{0,1\}$.

| Setting | 1 | 2 | 3 | 4 | 5 | 6 |
|---|---|---|---|---|---|---|
| **Censored Rate (%)** | 37.3 | 18.6 | 37.8 | 46.5 | 36.4 | 23.5 |
| **Treated Rate (%)** | 75.0 | 73.0 | 60.0 | 52.0 | 43.0 | 71.0 |
| $p(W=1,e=1)$ **(%)** | 47.3 | 60.6 | 38.0 | 32.8 | 38.9 | 55.3 |

## C  DATASET DETAILS

### C.1  SYNTHETIC DATASETS

In the six synthetic datasets used in Section 5.1, covariates $X$ are sampled from a uniform distribution $X\sim U[0,4]^P$, while the treatment variable is set as binary $\{0,1\}$ and determined by sampling from a binomial distribution $X\sim B(N,b)$, which indicates whether the treatment is administered. The base conditional survival time $T|X$:

$$T_{\mathrm{surv}}|X\sim\exp(\mathcal{N}(\mu(X),\sigma^2(X))),\tag{37}$$

where $\mathcal{N}(\mu(X),\sigma^2(X))$ is the normal distribution with mean $\mu(X)$ and standard deviation $\sigma(X)$ described in Table 2, along with dimension $p$ of $X$ and censoring time $C|X$ to mimic real-world censorship scenarios commonly encountered in clinical trials Zhou et al. (2020); Wilson et al. (2021). Furthermore, to simulate patients with random early mortality, we choose 5% of subjects, and reassign them the parameters $\mu=\ln 0.1$ and $\sigma=0.2$ instead, as utilized in Davidov et al. (2025). For the remaining 95%, the event time is defined as $T=T_{\mathrm{surv}}$.

In our work, under the SUTVA assumption, we aim to explore the differences in potential outcomes across different treatments. Therefore, while simulating the true censoring mechanism, the proportion of uncensored data is also crucial for our method, as shown in Table 3.

### C.2  REAL-WORLD CLINICAL DATASET

The dataset comes from a retrospective, single-institution cohort, comprising patients treated between January 2015 and June 2023. The institutional ethics committee has approved the usage of the dataset. The dataset includes 541 patients diagnosed with unresectable locally advanced non-small cell lung cancer (LA-NSCLC) who received radically intended radiotherapy (RT), with or without concurrent chemotherapy. Specific inclusion and exclusion criteria are detailed as follows:

- Inclusion criteria: 1) pathologically confirmed diagnosis of NSCLC; 2) age at least 18 years old; 3) KPS score at least 70; and 4) diagnosed as inoperable locally advanced NSCLC (AJCC 8th edition) before RT, with definitive RT.

Table 4: The ratio of different radiochemotherapy regimens in the clinical dataset analyzed in Section 5.2. For radiotherapy techniques, we have $0 :=$IMRT, $1 :=$VMAT, and for other chemotherapies $0 :=$False, $1 :=$True.

| Rate \ Radiochemo Therapy | Radiotherapy Techniques | Concurrent Chemotherapy | Induction Chemotherapy | Consolidation Chemotherapy |
|---|---|---|---|---|
| $p(W = 0)$ | 0.495 | 0.12 | 0.665 | 0.645 |
| $p(W = 1)$ | 0.505 | 0.88 | 0.335 | 0.354 |

Table 5: The value of six covariates in the clinical dataset analyzed in Section 5.2, where $\mathcal{X}_{-\text{lower}} = \{X_i : X_i \le \text{Quantile}(0.5, F_.)\}$, $\mathcal{X}_{-\text{upper}} = \{X_i : X_i > \text{Quantile}(0.5, F_.)\}$ and $F_.$ is the distribution of the corresponding covariate. The values of these six covariates are described below.

| Value \ Covariate | Stage | T-stage | N-stage | KPS | Max3D Diameter | Voxel Volume |
|---|---|---|---|---|---|---|
| 0 | IIB,IIIA | 1,2 | 0,1 | 70,80 | $\mathcal{X}_{\text{M3D}-\text{lower}}$ | $\mathcal{X}_{\text{VV}-\text{lower}}$ |
| 1 | IIIB,IIIC | 3,4 | 2,3 | 90,100 | $\mathcal{X}_{\text{M3D}-\text{upper}}$ | $\mathcal{X}_{\text{VV}-\text{upper}}$ |

- Exclusion criteria: 1) a history of lung surgery before RT; 2) prior chest RT; 3) poor quality of planning CT images; 4) unacceptable dose deviations; 5) treated with immunotherapy; and 6) a follow-up shorter than six months.

The overall survival (OS) is defined as the time from the start of RT to any cause of death or the last follow-up. A comprehensive set of variables was collected, including 17 clinical baseline characteristics (*e.g.*, age, gender, Karnofsky Performance Status, *etc*), tumor information (*e.g.*, pathology, staging, tumor location, *etc*), comorbidities, and treatment details. List of the 17 clinical baseline characteristics in the dataset

- Age $\in \{21, \cdots, 88\}$
- Gender $\in \{\text{Female}, \text{Male}\}$
- Karnofsky Performance Status (KPS) $\in \{70, 80, 90, 100\}$
- Pathology $\in \{\text{Squamous cell carcinoma}, \text{Non-squamous cell carcinoma}\}$
- Staging (AJCC 8th) $\in \{\text{IIB}, \text{IIIA}, \text{IIIB}, \text{IIIC}\}$
- T-stage $\in \{1, 2, 3, 4\}$
- N-stage $\in \{0, 1, 2, 3\}$
- Tumor Location $\in \{\text{Central}, \text{Peripheral}\}$
- Weight Loss $\in \{\text{NO}, \text{YES}\}$
- Hypertension $\in \{\text{NO}, \text{YES}\}$
- Diabetes $\in \{\text{NO}, \text{YES}\}$
- Smoking status $\in \{\text{NO}, \text{YES}\}$
- Radiation Dose ($\ge 60$Gy) $\in \{\text{NO}, \text{YES}\}$
- Radiation Technique $\in \{\text{IMRT}, \text{VMAT}\}$
- Induction Chemotherapy $\in \{\text{NO}, \text{YES}\}$
- Concurrent Chemotherapy $\in \{\text{NO}, \text{YES}\}$
- Consolidation Chemotherapy $\in \{\text{NO}, \text{YES}\}$

Additionally, the dataset includes 107 quantitative radiomic features of the tumor, for which the extraction method followed our prior work Luo et al. (2025). These features comprised 14 shape features, 18 first-order features, and 75 textural features. List of the 107 radiomic features extracted in the dataset:

Table 6: Configurations of the number of hidden layers of the MLP regressor implemented for the synthetic and real datasets.

| Setting | Hidden layers |
|---------|---------------|
| Synthetic dataset | $[8]$ |
| Clinical dataset | $[128, 64, 16]$ |

Table 7: Average runtime (seconds) based on the linear quantile regressor for each run comparison of different methods on simulated datasets, with 10 independent trials.

| Setting Method | 1 | 2 | 3 | 4 | 5 | 6 |
|---|---|---|---|---|---|---|
| Uncab | 0.024 | 0.029 | 0.048 | 0.043 | 0.064 | 0.220 |
| Naive | 0.416 | 0.483 | 0.991 | 0.832 | 1.26 | 6.76 |
| Focus | 0.212 | 0.234 | 0.329 | 0.291 | 0.397 | 1.16 |
| Fused | 10.89 | 12.81 | 21.1 | 13.09 | 25.07 | 58.6 |
| Ours | 1.32 | 1.42 | 1.52 | 1.51 | 1.72 | 3.13 |

- Shape features: Elongation, Flatness, LeastAxisLength, MajorAxisLength, Maximum2DDiameterColumn, Maximum2DDiameterRow, Maximum2DDiameterSlice, Maximum3DDiameter, MeshVolume, MinorAxisLength, Sphericity, SurfaceArea, SurfaceVolumeRatio, VoxelVolume.

- First-order features: 10Percentile, 90Percentile, Energy, Entropy, InterquartileRange, Kurtosis, Maximum, MeanAbsoluteDeviation, Mean, Median, Minimum, Range, RobustMeanAbsoluteDeviation, RootMeanSquared, Skewness, TotalEnergy, Uniformity, Variance.

- Textural features:

  - GLCM: Autocorrelation, ClusterProminence, ClusterShade, ClusterTendency, Contrast, Correlation, DifferenceAverage, DifferenceEntropy, DifferenceVariance, Id, Idm, Idmn, Idn, Imc1, Imc2, InverseVariance, JointAverage, JointEnergy, JointEntropy, MCC, MaximumProbability, SumAverage, SumEntropy, -SumSquares.

  - GLDM: DependenceEntropy, DependenceNonUniformity, DependenceNonUniformityNormalized,DependenceVariance,GrayLevelNonUniformity, GrayLevelVariance, HighGrayLevelEmphasis, LargeDependenceEmphasis, LargeDependenceHighGrayLevelEmphasis, LargeDependenceLowGrayLevelEmphasis, LowGrayLevelEmphasis, SmallDependenceEmphasis, SmallDependenceHighGrayLevelEmphasis, SmallDependenceLowGrayLevelEmphasis.

  - GLRLM: GrayLevelNonUniformity, GrayLevelNonUniformityNormalized, GrayLevelVariance, HighGrayLevelRunEmphasis, LongRunEmphasis, LongRunHighGrayLevelEmphasis, LongRunLowGrayLevelEmphasis, LowGrayLevelRunEmphasis, RunEntropy, RunLengthNonUniformity, RunLengthNonUniformityNormalized, RunPercentage, RunVariance, ShortRunEmphasis, ShortRunHighGrayLevelEmphasis, ShortRunLowGrayLevelEmphasis.

  - GLSZM: GrayLevelNonUniformity, GrayLevelNonUniformityNormalized, GrayLevelVariance, HighGrayLevelZoneEmphasis, LargeAreaEmphasis, LargeAreaHighGrayLevelEmphasis, LargeAreaLowGrayLevelEmphasis, LowGrayLevelZoneEmphasis, SizeZoneNonUniformity, SizeZoneNonUniformityNormalized, SmallAreaEmphasis, SmallAreaHighGrayLevelEmphasis, SmallAreaLowGrayLevelEmphasis, ZoneEntropy, ZonePercentage, ZoneVariance.

  - NGTDM: Busyness, Coarseness, Complexity, Contrast, Strength.

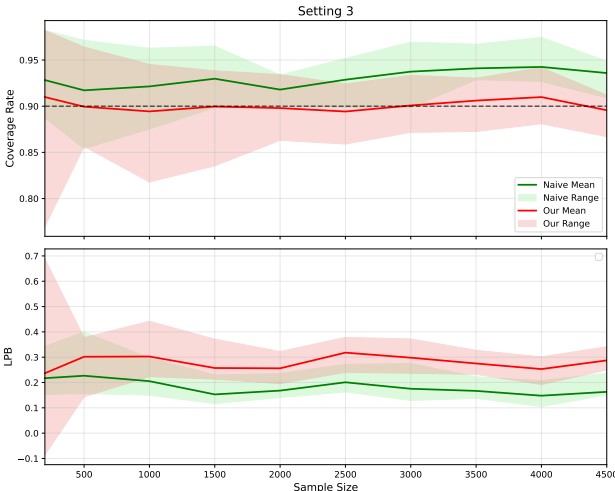

Figure 6: Performance of the naive and our methods as a function of sample size in Setting 3. **Top**: empirical coverage rate, with a red dashed line indicating the nominal 90% level. **Bottom**: LPB. A higher LPB is better. The performance metrics are evaluated on 10 independent trials, each consisting of newly sampled train, validation, calibration, and test sets of ratios 50%, 10%, 30%, and 10% of the synthetic dataset with different sample sizes, respectively.

## D   EXPERIMENT SETUP

The data is split into four folds in all experiments: 50% for training, 30% for calibration, 10% for validation (used for early stopping), and 10% for testing to evaluate performance. As described in Table 2, the synthetic data is generated through distribution simulations. The collection and inclusion of the clinical dataset are described in Appendix C.2.

In all experiments, we estimate the counterfactual quantile regression function by using an MLP neural network implemented in PyTorch based on CQRNN Pearce et al. (2022). The MLP regressor consists of three hidden layers with ReLU activation, early stopping (triggered after 5 epochs without improvement), and a training cycle of 400 epochs. The SGD optimizer is used to optimize the model with parameters learning rate = $1e-3$, weight decay= $1e-4$, and momentum= $0.9$, a batch size of 256, dropout layers with a rate of $p = 0.1$, and varying configurations of hidden layers, detailed in Table 6. Besides, we employ *scikit-learn* Pedregosa et al. (2011) to train a Random Forest Classifiers with maximum depths of 10 and 4, to estimate the weight function $\gamma(x)$ for real data and synthetic ones, respectively. The lower max depth in the synthetic experiment is for reducing potential overfitting.

### D.1   DETAILS OF MULTI-TREATMENT ON SETTING 4.

We randomly divide the data into three parts with the same ratio. Based on the data generation process of Setting 4, we add Gaussian perturbations of $\mathcal{N}(0.5, 0.2)$ and $\mathcal{N}(1.0, 0.2)$ to the data under Treatment 1 and Treatment 2, respectively. The experiment aims to simulate the differential effects of various treatments on survival time. The results show that our method can distinguish the distinct impacts from different treatments and provide informative LPB.

### D.2   DETAILS OF SIMULATION ON ABOUT EXTREME CASES.

The extreme cases (outliers) are added into setting 4 as shown in Figure 3, to simulate real-world scenarios where conditions such as interstitial lung abnormalities (ILA) Kashihara et al. (2023) and untreated chronic obstructive pulmonary disease (COPD) Jo et al. (2022) can lead to drastically reduced patients' survival times during treatment. In practical terms, we introduce perturbations to the survival times for 10% of the data randomly selected from the dataset generated under Setting

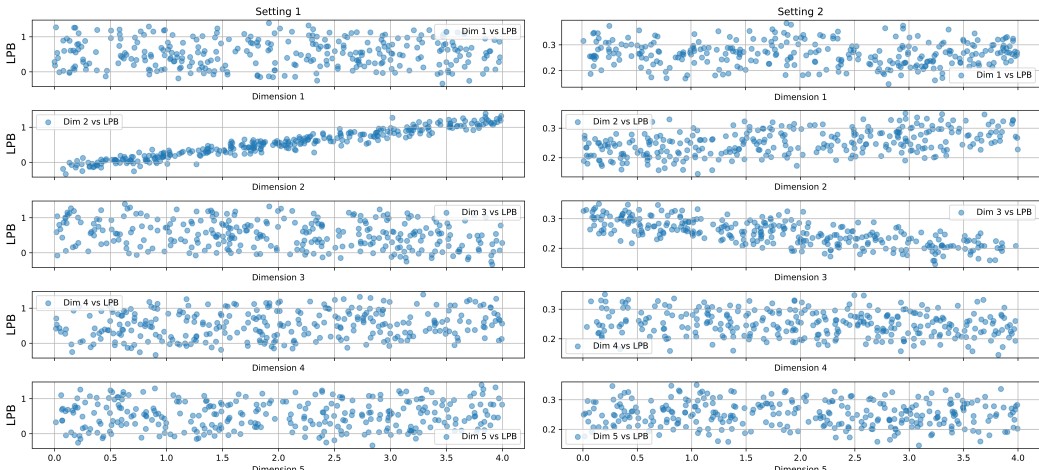

Figure 7: LPB as a function of different covariates in the synthetic datasets of Setting 1 and Setting 2, with a total sample size of 3000, with 10% of the data for testing.

4, by subtracting Gaussian noise sampled from $\mathcal{N}(1,2)$, $\mathcal{N}(10,2)$, and $\mathcal{N}(20,2)$, respectively. The perturbed values are then truncated at zero by taking the maximum with zero.

### D.3 MACHINE SPECIFICATION

The hardware and operating system used for the experiments are as follows.

- **CPU:** Intel(R)Xeon(R) Platinum 8369B CPU @ 2.90GHz
- **GPU:** NVIDIA A100-80G-SXM
- **System:** Ubuntu 22.04

### D.4 COMPUTATIONAL EFFICIENCY

In addressing the covariate shift from $\mathbb{P}_{X|W=w,e=1}$ to $\mathbb{P}_X$, it requires the computation of the non-conformality score $V_i^{(w)} = \hat{q}_\tau^{(w)}(X_i) - \widetilde{T}_i$ within $\mathcal{D}_{\text{cal}}$ and finding the weighted $(1-\alpha)$-th quantile, resulting in a computational complexity of $O(|\mathcal{D}_{\text{cal}}|)$ for the calibration process. The remaining computational overhead lies in the training procedures of the counterfactual regression function $\hat{q}_\tau^{(w)}(x)$ and the weight function $\hat{\omega}(x)$. The runtime comparison is shown in Table 7, the runtime increases with the increase in dimension. The runtime of the fused method is the longest due to its implementation, which results in three predictions from the weight function for each calibration data point. Our method demonstrates a computationally efficient property of runtime: (comparable to the naive method, and faster than the fused method) while providing strong theoretical guarantees and robust performance in challenging scenarios (e.g., multi-treatment, extreme cases). The "naive calibration process" does not significantly increase runtime, demonstrating that our method is practical for real-world applications—it can deliver accurate and robust results without compromising speed.

## E ADDITIONAL EXPERIMENT

In addition to comparisons with other baseline methods, we conduct several additional experiments on synthetic datasets to demonstrate the property of our conformalized survival counterfactuals prediction procedure between the naive method and ours. In Section E.1, we explore the impact of sample size on performance. And we examine the adaptiveness of our method in Section E.2. Then, in Section E.3, we investigate whether the method's performance is affected by $p(W = w, e = 1)$ within a reasonable range in the dataset. The experiment of different regression algorithms for our method is provided in Section E.4. Finally, we compare the baselines with coverage guarantee to our method in real data in Section E.6.

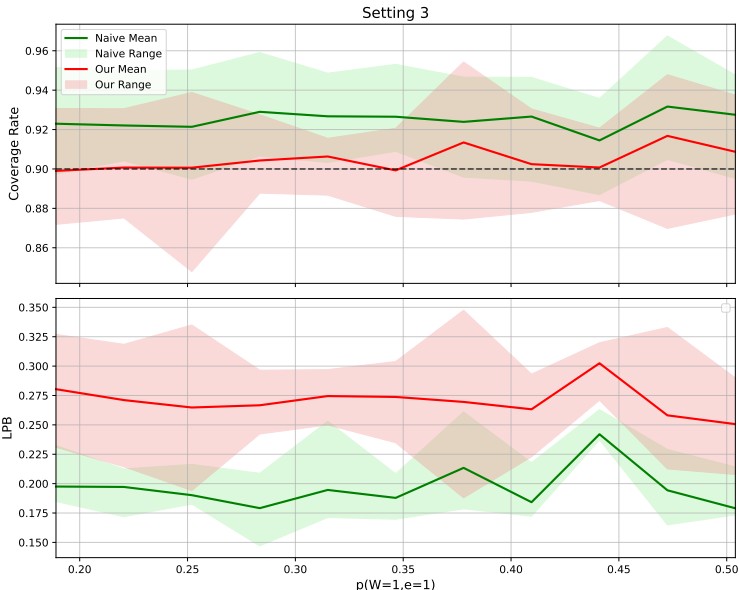

Figure 8: Performance of the naive and our methods as a function of $p(W = 1, e = 1)$ in Setting 3. **Top**: empirical coverage rate, with a red dashed line indicating the nominal 90% level. **Bottom**: LPB. A higher LPB is better. The performance metrics are evaluated on 10 independent trials, each consisting of newly sampled train, validation, calibration, and test sets with ratios of 50%, 10%, 30%, and 10% of the synthetic datasets, with a total sample size of 3000, respectively.

## E.1 EFFECT FROM THE SIZE OF DATA

In this section, we examine both the coverage rate and LPB with varying total sample sizes of the dataset, increasing from 200 to 500, then to 4500 in intervals of 500. The results shown in Figure 6 indicate that our method, based on the SUTVA and strong ignorability assumptions, utilizes a part of training and calibration data in practice, making it more sensitive to changes in data volume compared to the naive method when the data volume is small. When the total sample size is below 1000, our calibration strategy demonstrates less robustness to the effect from sample size compared to the naive method, while the interval of LPB shows a marked reduction when the sample size exceeds 500. Both methods stabilize after the total data volume exceeds 2000, and our method achieves a tighter LPB with a valid coverage rate.

## E.2 ADAPTIVENESS ANALYSIS

In this section, we explore the adaptiveness of our method across different covariates. As shown in Figure 7, the LPB of our method demonstrates significant variations corresponding to changes in $x_2$, and in both $x_2$ and $x_3$, on Settings 1 and 2, respectively. Similarly, we also investigate the adaptiveness of the method on real clinical data, with the results presented in Section 5.2. The results show that our method exhibits data adaptability both in synthetic and clinical data for providing a reliable LPB for survival counterfactuals prediction.

## E.3 EFFECT FROM THE RATIO OF $p(W = w, e = 1)$

Based on the derivation in (3), the theoretical performance of our method remains invariant to changes in $p(W = w, e = 1)$. In this section, we investigate the changes in empirical coverage rates and LPB of the naive method and our method under Setting 3 ($W \in \{0, 1\}$) when $p(W = 1, e = 1)$ varies within a reasonable range. In the simulation experiment, we maintain the censoring rate on Setting 3 and vary the treatment ratio $p(W = 1)$ from 0.2 to 0.5 in increments of 0.05 by adjusting the parameter of the binomial distribution. The results, shown in Figure 8, demonstrate that both methods achieve valid coverage rates across the reasonable range of $p(W = 1, e = 1)$ variations, while our method produces less conservative LPB.

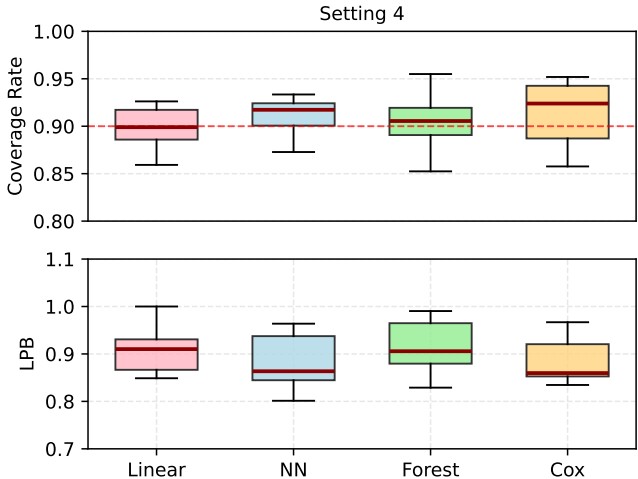

Figure 9: Different quantile regressors, linear regression (Linear), neural network (NN), random forest regressor (Forest), and Cox model (Cox), are utilized in our method on setting 4 with 10 independent trials.

### E.4   THE CHOICE OF DIFFERENT REGRESSION ALGORITHMS.

Based on setting 4, we apply four different quantile regressors, namely linear regression (Linear), neural network (NN), random forest regressor (Forest), and Cox model (Cox), with the weight function being a random forest, as shown in Figure 9. The results indicate that our method achieves the coverage guarantee and exhibits low sensitivity across the four different regression algorithms evaluated.

### E.5   THE CHOICE OF DIFFERENT WEIGHT FUNCTIONS.

We explore four different classifiers for the weight function, namely random forest (Forest), logistic regression (Logistic), neural network (NN), and support vector machine (SVM), with the regressor being a neural network, as shown in Figure 12. The results indicate that our method achieves robustness with different weight functions, both for the coverage rate and LPB.

### E.6   COMPARISON OF BASELINES ON REAL DATA.

To evaluate the performance of different methods on real-world data, we conducted a four-group experiment comparing Radiotherapy Techniques (IMRT vs. VMAT) and Consolidation Chemotherapy (True vs. False). The methods achieving the desired coverage guarantee are compared and illustrated in Figure 10. The results demonstrate that each method can effectively distinguish the impact of IMRT vs. VMAT and the presence or absence of Consolidation Chemotherapy on survival time. Notably, our method not only satisfies these conditions but also provides the most informative LPB.

## F   THE USE OF LARGE LANGUAGE MODELS (LLMS)

We declare that in this work, LLMs were used solely for grammatical correction in writing.

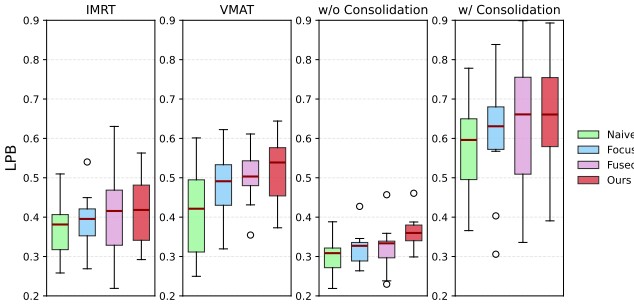

Figure 10: LPB comparison from the baselines with coverage guarantee of $\alpha = 0.1$ for different treatment regimes, with 10 independent trials.

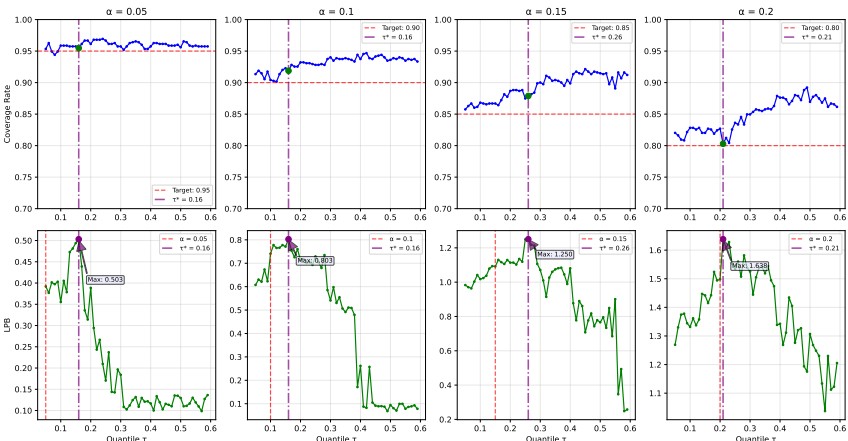

Figure 11: The variation of the LPB provided by our method with different $\tau$ of quantile regression under various $\alpha$ settings.

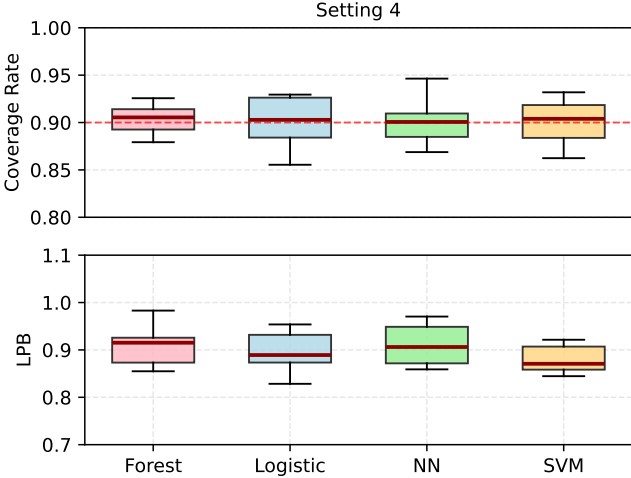

Figure 12: Different weight functions (classifier), random forest (Forest), logistic regression (Logistic), neural network (NN), and support vector machine (SVM), are applied in our method on setting 4, with 10 independent trials.

