# OpenReview forum: "Conformalized Survival Counterfactuals Prediction for General Right-Censored Data"
_ICLR.cc/2026/Conference — ICLR 2026 Poster_

### Official Review · Reviewer_dYEk · 2025-10-28

**Soundness:** 3
**Presentation:** 3
**Contribution:** 3
**Rating:** 8
**Confidence:** 3

**Summary:**

The paper introduces a novel approach to do conformalized survival counterfactuals prediction, which mainly has two advantages: 1) provide valid LPB for counterfactual survival outcomes for general right-censored data; 2) achieve the marginal coverage with the double robustness property instead of the PAC-type guarantee, which should be more reliable when rare and extreme cases are present.

The author proposes a new non-conformity score based on the quantile regression and use the probability $P(V(X, \tilde{T})\leq c_{1-\alpha}(\tau))$ with the help of weight function $w(X)$ to learn the cut-off value, which allows for inference on general censored data with exact coverage guarantee compared to the use of empirical version for $\alpha(\tau)$ .

**Strengths:**

1. transform the problem into a weighted conformal inference problem with exact coverage for the entire population.
2. LPB optimization to get the highest LPB with the coverage guarantee.
3. doubly robust with theoretical guarantee if the weight function is estimated rather well.

**Weaknesses:**

* The definition of $c_{1-\alpha}^{(w)}(\tau)$ appears to be incorrect, as the right-hand side should also depend on $w$. For example, it should be a function of $V^{(w)}(X, T\leq c)$.

* It would be beneficial to provide the selected $\tau$ in the LPB optimization. I suspect that the optimal $\tau$ should be close to $0.5$, as this would allow the majority of the data to be leveraged for a more accurate estimation of $q_{\tau}(x)$.

* If the extreme cases occur with very small probability, achieving exact marginal coverage should intuitively inflate the conformalized interval (i.e., result in a lower LPB). I did not observe this effect in Figure 1. Is this because the data generation process for the synthetic datasets does not produce too extreme values, or is it an artifact of the LPB optimization step? I believe an ablation study on the LPB optimization would be a valuable addition to the paper, as it would experimentally highlight the importance of achieving exact coverage.

**Questions:**

None

---

> ### Author Response · Authors · 2025-11-21
>
> Thank you for your efforts in reviewing our paper. We appreciate the recognition of the novelty of our method and theoretical analysis. Our response is provided below.
>
> **Comment:** About experiment with different $α$.
>
> **Response:** In our revised paper, we have reported the LPB and coverage rate for our optimization procedure across different $α$ values in Table 1 and Figure 11. The result shows that the LPB given the optimal $τ$ is comparable to that by setting $τ = α$, suggesting that our quantile regression estimator is well trained.
>
> **Comment:** The strength of our method in handling extreme cases.
>
> **Response:** Thanks for your suggestions. We verify the robustness of our methods and baselines against outlier samples in Figure 3. With outlier data, our method can consistently achieve the desired coverage, while those with only PAC-type coverage fail to do so. These results align well with our motivation and theoretical claims.

---

### Official Review · Reviewer_xp44 · 2025-10-30

**Soundness:** 2
**Presentation:** 2
**Contribution:** 2
**Rating:** 4
**Confidence:** 2

**Summary:**

This paper proposes a method to estimate the lower prediction bound for the survival time of different treatments under general right-censored data. For that, the authors leverage ideas from weighted conformal prediction in the survival setting under standard assumptions of the potential outcomes framework. The authors provide theory for coverage guarantees and a double robustness property of their framework. Lastly, they perform experiments on synthetic data to compare their method against baselines, and provide results on a real-world medical dataset to show the applicability of their method in practice.

**Strengths:**

- Uncertainty quantification for counterfactual prediction of survival data is an important problem for real-world medical treatment settings and using conformal prediction seems to be an useful research direction here.
- The authors provide results on real world data sets and give insights and evaluation of the plausibility of their estimates.

**Weaknesses:**

Some parts of the theoretical contribution seem unclear to me and the comparison to prior related work could be improved (however, I do not have further background knowledge on conformal prediction for survival data).
- Especially the contribution over Candes (2023) is not clear to me (see also Questions).
- E.g., in ll. 182, the authors state, unlike previous methods, their method would achieve exact marginal coverage. However, Theorem 4.1 shows that the method only achieves marginal coverage when $\hat{w}(x) = w(x)$ if I understand it correctly? Could the authors elaborate on this?

The experimental results section could be improved especially with respect to clarity and presentation.
- At least a short summary of the used datasets and a bit more details should be included in the main paper as this is relevant to check the robustness of their method (simple vs complex setup) and practical relevance (real-world applicability, and realistic setup?).
- the baselines should be cited and described shortly again in the experiments to give intuition about their potential shortcomings.
- The used metrics (empirical coverage, relative LPB) should be at least once defined formally.
- The presentation of the results could be improved. In Figure 1, especially the lower row is not easy to interpret. The authors state in the caption of Fig. 1, a higher LPB would be better but write in the text uncab (with highest LPB) would produce a conservative LPB estimate, this seems inconsistent to me. I guess, overall, the authors would like to display a tradeoff between valid empirical coverage and less conservative/tighter LBP estimates? One possible way to show this would be to display both in one graph with coverage and LPB on the axis and show this for over different values for $\alpha$ to show that their method dominates the others?
No code is provided for reproducibility.

Minor:
- citation style needs to be adjusted (direct vs indirect citations)
- In the introduction the double robustness property is mentioned. It should be summarized once in the beginning what this means and double robustness with respect to which nuisance functions is meant.

**Questions:**

- The authors mention “Candes et al. (2023) also provided survival counterfactuals prediction, but only used on cases with censoring time exceeding a specific threshold.” Could the authors elaborate on the difference to that and what the implications of this shortcoming are? That would help to stress their contribution more clearly.
- For the real-world experiments, how can the empirical coverage given the different treatment strategies be evaluated without access to ground-truth counterfactuals/survival times?
- For the real-world experiments, how do the results and findings of the authors’ method deviate or coincide with the baselines?

---

> ### Author Response · Authors · 2025-11-21
>
> Thank you for your feedback and efforts in reviewing our paper. We address your questions below.
>
> **Comment: the novelty of our method from Candes et al. (2023).**
>
> **Response:** [1] focused on the Type-I right-censored data, which assumes that the censoring time $C_i$ is known, and its calibration relies on the availability of $C_i$. In contrast, we considered the counterfactual prediction in the general right-censored setting, where $C_i$ is not known.
>
> **Comment: achieve marginal coverage when $\widehat{\omega}(x) = \omega(x)$.**
>
> **Response:** Yes, please refer to Proposition 1 in [2], where they assume $\widehat{\omega}(x) = \omega(x)$ to achieve exact coverage. Here, we use "exact" to contrast with the "approximate" PAC-type coverage, which only guarantees coverage with high probability over the training data.
>
> In practice, as long as $\widehat{\omega}(x)$ is $L_1$-consistent (assuming correct specification) and the sample size is sufficiently large, the resulting error becomes negligible. Furthermore, Theorem 4.2 establishes the doubly robustness property: as long as either $\widehat{\omega}$ or $q$ is correctly specified, the coverage error remains negligible.
>
> **Comment: details about real data.**
>
> We have introduced the real dataset briefly at the beginning of Section 5.2, with details left in Appendix C.2.
>
> **Comments: comparisons with baselines in real data.**
>
> Thanks for your suggestion. In our revised version, we have compared with baselines that achieved coverage guarantee in numerical data, and the results is present in Figure 10. Our procedure can produce higher LPB, suggesting its informativeness in predicting the survival time.
>
> **Comments: analysis about Figure 1.**
>
> While the uncab method achieves the highest LPB, it does not satisfy the coverage guarantee. As claimed in lines 128-131, our goal is to find a valid LPB with desired coverage. On this basis, we further optimize lines 251-259 to maximize the LPB, making it as informative as possible.
>
> **Comment: evaluation without access to ground-truth counterfactuals/survival times.**
>
> **Response:** Unlike prediction or regression tasks, where test data have ground truth, statistical inference/causal inference typically lacks ground truth in real data. Consequently, evaluation usually relies on interpreting results in light of previous studies and expert knowledge. In our paper, our findings are consistent with prior research.
>
> **References**
> [1] Candès, E. J., & Ren, Y. (2023). Conformalized survival analysis. Journal of the American Statistical Association.
> [2] Lei, J., & Candès, E. J. (2021). Conformal inference for individual treatment effects. Journal of the Royal Statist.

---

### Official Review · Reviewer_Mprf · 2025-10-31

**Soundness:** 3
**Presentation:** 3
**Contribution:** 3
**Rating:** 6
**Confidence:** 3

**Summary:**

This paper aims to give lower prediction bounds (LPBs) for counterfactual survival time under different treatments with general right-censoring. While prior conformal survival methods gave PAC-type guarantees; this work targets marginal population coverage for counterfactual survival times, under the strong ignorability assumption.

The method adopts the weighted conformal prediction framework. Experiments on synthetic and a 541-patient NSCLC lung-cancer data show near-nominal coverage and informative bounds.
Theory gives a finite-sample lower bound on marginal coverage that degrades with density-ratio error, and a doubly robust asymptotic result if either the weight estimator or the quantile estimator is consistent.

**Strengths:**

1. Important problem: The paper tackles a meaningful and practical challenge—making counterfactual survival time prediction reliable when data are right-censored. This is essential for treatment-effect estimation in medicine, where censoring is unavoidable.

2. Solid theoretical foundation: The authors derive finite-sample marginal coverage guarantees and show that coverage error depends only on the estimation error of a single density ratio.

3. Consistent empirical performance: The method performs reliably on six diverse synthetic setups and on a real high-dimensional clinical dataset, showing both strong coverage and clinically plausible counterfactual predictions.

**Weaknesses:**

1. Coverage depends on ratio accuracy: The claim of “exact” coverage only holds when the density ratio is estimated perfectly. Theorem 4.1 actually provides a lower bound that subtracts an error term based on the ratio estimation error, so the language precision may be worth reviewing

2. Limiting assumptions: The method assumes strong ignorability, meaning the potential outcomes are independent of both treatment and censoring once covariates are conditioned on. This can be a demanding assumption in practice.

3. Narrow empirical validation: Experiments only test a single coverage level ($\alpha=0.1$). Broader evaluation would make the empirical evidence more convincing.

**Questions:**

1. Could you please provide some results for for other values of $\alpha$ to show consistency of the performance?

2. Compare to previous conformal survival works, is it true that one reason the theory successfully avoids the PAC-style guarantee is because of the survival counterfactuals setting. Could you please provide some intuition for why this joint setting enables that?

---

> ### Author Response · Authors · 2025-11-21
>
> Thank you for your positive feedback regarding the theoretical contribution and empirical validation. Our response is provided below.
>
> **Comment:** The terminology of "exact guarantee" and the reason avoid PAC-type guarantee and the practical value of our procedure.
>
> **Response:** Here, we use "exact" to contrast with the "approximate" PAC-type coverage, which only guarantees coverage with high probability over the training data. The exact coverage can be achieved when $\widehat{\omega}(x) = \omega(x)$, which is also achieved in Proposition 1 in [1]. Indeed, this error can be negligible when $\widehat{\omega}$ is correctly specified and the sample size is large enough.
>
> **Comment:** About the limiting assumption (strong ignorability) in our method.
>
> **Response:** The ignorability assumption is standard in conformal prediction for causal inference, as commonly adopted in [1,2]. While relaxing this assumption is important, it is not the focus of our paper. Indeed, sensitivity analyses like those in [3] could be integrated into our approach, but this exploration is independent of our main contributions.
>
> **Comment:** Exact coverage is due to the setting considered in this paper.
>
> **Response:** No. In fact, achieving exact coverage is more challenging in the general right-censored setting, since the censoring time $C_i$ is not given in our case, whereas it is given in the type-I right-censored setting [2]. Works such as [4,5] consider the same setting as ours but only achieve PAC-type coverage.
>
> **Comment:** About the experiment with different $α$.
>
> **Response:** We have provided results for different $α$ in Table 1 in the revised manuscript. The result suggests that our method provides the coverage guarantee for different $α$ within a reasonable range.
>
> **References**
> [1] Lei, J., & Candès, E. J. (2021). Conformal inference for individual treatment effects. Journal of the Royal Statistical Society.
> [2] Candès, E. J., & Ren, Y. (2023). Conformalized survival analysis. Journal of the American Statistical Association.
> [3] Jin, Y., & Candès, E. J. (2023). Sensitivity analysis for conformal inference. Biometrika.
> [4] Gui, Z., & Liu, R. (2024). Conformalized quantile regression for survival data. Biometrics.
> [5] Davidov, O., & Fink, L. (2025). Conformal prediction for right-censored data. Journal of Machine Learning Research.

---

### Official Review · Reviewer_rUA7 · 2025-11-01

**Soundness:** 3
**Presentation:** 3
**Contribution:** 2
**Rating:** 6
**Confidence:** 4

**Summary:**

The proposed CSC method develops a weighted conformal calibration procedure under the potential outcome framework, producing exact marginal coverage and a doubly robust property against model misspecification. Theoretically, the authors derive formal coverage guarantees and empirically validate the approach on synthetic simulations and a real lung cancer dataset.

**Strengths:**

1. The authors provide formal proofs for marginal validity and robustness.
2. Introducing conformal calibration for counterfactual survival with exact coverage is a meaningful advance.
3. Enhances credibility and robustness under imperfect model estimation.

**Weaknesses:**

1. The experiments, though well organized, are limited to one real-world dataset (a single lung cancer cohort). This constrains the generalizability of the findings. Additional datasets from other clinical domains or public survival benchmarks (e.g., SUPPORT, METABRIC, MIMIC, SEER) would strengthen the empirical evidence.
Moreover, ablation studies isolating the contributions of key components—such as the choice of quantile regression model (MLP vs. CQR forest), weighting scheme, and calibration size—are missing. Such ablations would help quantify where performance gains arise.

2. The proposed method introduces several estimation layers: quantile regression, density ratio estimation ($\hat{\gamma}(x)$), and weighted conformal calibration. These can be computationally intensive and may suffer from instability when sample sizes are small or censoring rates are high. The paper does not analyze runtime complexity, convergence stability, or the effect of hyperparameter choices. In large-scale or high-dimensional settings, density ratio estimation can be error-prone and potentially undermine the theoretical guarantees if $\hat{\omega} (x)$ is poorly estimated.

3. The paper mainly compares against a few conformal survival calibration baselines (uncalibrated, naive, focused, fused). However, it omits other relevant conformal-based approaches that are published in recent years. Inclusion of such baselines would clarify whether CSC’s advantages stem from conformalization or from model choice.
Additionally, it would be useful to report not only coverage and LPB but also metrics like mean absolute deviation, average interval width, and computational cost for a fuller comparison.

**Questions:**

1. Could the authors provide runtime comparisons with existing conformal survival approaches?

---

> ### Author Response · Authors · 2025-11-21
>
> Thank you for your recognition of the significance of the proposed problem, and the novelty of our method. Our response is listed below.
>
> **Comment:** The reason we chose the real data collected from the hospital recently, rather than some other public datasets.
>
> **Response:** We thank the reviewer for the constructive suggestion. Our real-world experiment focuses on a lung cancer cohort because the goal is to evaluate the proposed method in a clinically meaningful setting, where detailed covariates, censoring patterns, and domain-specific structures are available. We believe this dataset is sufficiently rich to demonstrate the practical value of our approach. In contrast, existing benchmarks have missing information. Specifically, TCIA lacks survival follow-up, and other data lack detailed treatment parameters that are key for modeling survival prediction. Additionally, our extensive synthetic and semi-synthetic studies already examine diverse settings—covering varying censoring rates, hazard structures, model misspecification, and distribution shifts—providing strong evidence of generalizability. Evaluating additional clinical cohorts is a valuable extension, and we plan to include such analyses in future work.
>
> **Comment:** Ablation study about sample size, regressors, $\tau$, and weight function.
>
> **Response:** Thanks for your suggestions. We will explain these estimation components one by one.
>
> - **About sample size:** In our original version, we have verified the effect of sample size in Figure 4 of the appendix. The results indicate that our method is robust to changes in sample size.
> - **About quantile regression estimator:** In Figure 9 of the revised paper, we evaluated different models for fitting the quantile regression, and the results demonstrate that our method consistently achieves the desired coverage across all choices.
> - **About hyperparameter $\tau$:** We have performed the sensitivity analysis with different $\tau$ in quantile regression in Table 1 and Figure 11 of our revised paper. As shown, we have desired coverage in the full spectrum of $\tau$ while the optimal $\tau^*$ selected by our optimization algorithm achieves the highest (most informative) LPB.
> - **About the weight function:**  We have added the sensitivity analysis about different weight functions, and the results show that our method consistently achieves the desired coverage across all choices. The results are shown in Figure 12 and Appendix E.5 of the revised manuscript
>
> **Comment:** Comparisons with more baselines.
>
> **Response:** While many conformal methods have been developed in recent years, only a few focus on counterfactual prediction in survival analysis under the general right-censored setting. We compare our method with Focus and Fused [2], two recent and representative approaches in this setting.
>
> **Comment:** Run time analysis for our method.
>
> **Response:** Thanks for your suggestion. We have reported the running time of our method and the baseline models in Table 7, with a detailed analysis provided in Appendix D.4 of the revised manuscript. The result suggests that our method can deliver accurate, robust results without compromising on speed.
>
> **References**
> [1] Candès, E. J., & Ren, Y. (2023). Conformalized survival analysis. Journal of the American Statistical Association.
> [2] Davidov, O., & Fink, L. (2023). Conformal prediction for right-censored data. Biometrics.

---

> > ### Comment · Reviewer_rUA7 · 2025-11-27
> >
> > I appreciate the author’s rebuttal, especially the inclusion of many (ablation and running time) experiments. Most of my questions have now been addressed. My only remaining suggestion is to add at least one additional dataset (it doesn’t need to be public if clinical insight is lacking). Overall, I think the paper is decent.

---

> ### Author Response · Authors · 2025-11-28
>
> We want to express our sincere gratitude for your valuable suggestions, which have significantly enhanced the completeness of our work. We will continue to make improvements.

---

### Official Review · Reviewer_2PvE · 2025-11-05

**Soundness:** 2
**Presentation:** 2
**Contribution:** 2
**Rating:** 2
**Confidence:** 4

**Summary:**

The paper proposes a new method to provide lower prediction bounds for counterfactual outcomes of binary treatments in right-censored survival data based on conformal prediction. In contrast to the former work, which only provides approximate guarantees, the proposed method provides exact marginal coverage guarantees. The core idea of the method is based on a transformation of the coverage probability into a reweighted expectation.

**Strengths:**

- The paper addresses an interesting and important topic: Finite sample uncertainty quantification in counterfactual prediction in time-to-event data.
- The paper provides rigorous mathematical guarantees and derivations.

**Weaknesses:**

- The paper lacks a proper related work discussion on conformal prediction for causal effects.
- In the motivation, the paper states that it provides bounds for "different treatments". However, later on, the treatment is considered binary, rendering the former an overstatement.
- The paper combines existing work in a straightforward manner. I do not see much novelty but rather a direct application of Lei and Candès (2021) and Candès et al (2023).
- The work contains many wrong statements (potentially due to a last-minute submission). For example, line 188 states that the upper bound exactly equals alpha. This is not true. In line 74, the paper (presumably) confuses upper and lower bounds.
- The paper requires the weight function to be well estimated. If it is not, the coverage guarantee could even be 0. This is not useful in practice.
- The experimental evaluation is insufficient to assess the benefit of the proposed method over, e.g., the fused method by Davidov et al. (2025). Statements made on the comparison are not necessarily true. A significantly larger lower bound is not observable in the stated settings 3,4,5.
- The paper is not yet ready for publication. Besides an often unstructured presentation with confusing mistakes and unexplained mathematical notation, this can be seen in the incorrect specification of the references, e.g., journal name "find this article online" or incomplete conference proceedings "proceedings of the... conference."

**Questions:**

- Lines 141 following: what is $\tau$? The notation has not been introduced.
- Counterfactual quantile regression (lines 168 following): What exactly is the relation between $\tau$ and $\alpha$?
- Double robustness provides asymptotic guarantees in contrast to the finite-sample CP guarantees. Why does it make sense to assess the double robustness of the CP interval? What does it mean in practice? Which value does Theorem 4.2 provide for the CP intervals?
- Experiments: how does the variance of the coverage and the lower bound compare to other methods? This is not observable from the current evaluation. How does the method perform over different levels of alpha? In comparison to the fused method, one can observe, on average, a larger variance in coverage and a lower bound. This does not support the effectiveness of the proposed method.

---

> ### Author Response · Authors · 2025-11-21
>
> Thank you for reviewing our paper. Our responses are listed below. We have corrected the formatting errors in the references and addressed the grammatical mistakes accordingly.
>
> **Comment:** Discussion of related works.
>
> **Response:** We have discussed the related works about conformal prediction in counterfactual and individual treatment effects in lines 98-103.
>
> **Comment:** Conformal calibration across "different treatments".
>
> **Response:** While we only evaluated two treatments (IMRT and VMAT) in a real-world experiment, it can be trivially extended to multiple treatment scenarios. This is because our calibration does not rely on the number of treatments. As seen in line 130 and line 281, our target and calibration is applicable to any treatment $ω$.
>
> **Comment:** Novelty of our method compared to Lei and Candes (2021) and Candes et al (2023).
>
> **Response:** Our method does not use the one in [1], and only employ the method in [2] in the last step. Indeed, [1] and [2] consider settings that differ from ours and are not directly applicable to our settings. Specifically, [2] focuses on individual treatment effects, while [1] addresses survival analysis under a Type-I censoring setting, which assumes that the censoring time $C_i$ is known. In contrast, our work tackles the more general right-censored setting. Moreover, the calibration procedure in [1] heavily relies on the availability of $C_i$, as detailed in Section 2.3 of their paper.
>
> **Comment:** Estimation of weight function, its influence on the coverage, and doubly robustness of our method.
>
> **Response:** Estimating the weight function and performing quantile regression are well-established in the literature [2,3,4]. Additionally, Theorem 4.1 accounts for the calibration error, while Theorem 4.2 demonstrates the doubly robust property of our procedure. Doubly robustness is a basic concept in statistical inference and machine learning, ensuring reliable performance even if one of the two models is misspecified. In our context, these models refer to the quantile regression estimator and the weight function, and this property is a key feature in our work as well as prior works [2,3].
>
> **Comment:** Results comparison from simulation of different settings to the method from Davidov et al. (2025).
>
> **Response:** In settings 3, 4, and 5, the box lengths in the Box-Plot of Figure 1 indicate that the median coverage rates of our method are closer to the desired values compared to the fused method. Additionally, our method shows fewer outliers in its LPBs than the fused method.
>
> To further verify the robustness of our method against outlier data, we also compare these methods in terms of coverage rate in Figure 3 in the revised version. As shown, our method consistently achieves coverage guarantee while those methods with only PAC-type coverage fail to do so. These results can justify our motivations and theoretical claims.
>
> **Comment:** Definition of $τ$ from $q̂_τ$ first time appearance from Gui et al. (2024) in the chapter preliminary.
>
> **Response:** We have claimed in line 167 that $q_τ(x)$ is the $τ$-th quantile of $T|X=x$, such that $q_τ(x) :=$ inf $\lbrace t: P(T ≤ t|X=x) ≥ τ \rbrace$ (line 180) or equivalently sup$\lbrace t: P(T ≤ t|X=x) ≤ τ \rbrace$.
>
> **Comment:** The relationship between $τ$ from $q^{(w)}\_τ (x)$ and $α$ from $L^{(w)}\_{α}(x)$.
>
> **Response:** In our method, $α$ and $τ$ are unrelated. $α$ is the fixed target for the miscoverage rate, representing the desired level of error in coverage. On the other hand, $τ$ is used to define the conditional quantile $q_τ$, and it can take any value depending on the quantile of interest. We introduce $τ$ to avoid confusion with $α$, as $α$ is fixed, while $τ$ is a free parameter that can vary. Therefore, $α$ controls the miscoverage rate, while $τ$ specifies the quantile, ensuring clarity in the notation.
>
> **Comment:** Statement of the upper bound of the miscoverage rate.
>
> **Response:** Thank you for your suggestion. We apologize for these typos and have corrected them in the updated version.
>
> **References**
> [1] Candès, E. J., & Ren, Y. (2023). Conformalized survival analysis. Journal of the American Statistical Association.
> [2] Lei, J., & Candès, E. J. (2021). Conformal inference for individual treatment effects. Journal of the Royal Statistical Society.
> [3] Gui, Z., & Liu, R. (2024). Conformalized quantile regression for survival data. Biometrics.
> [4] Davidov, O., & Fink, L. (2025). Conformal prediction for right-censored data. Journal of Machine Learning Research.

---

> > ### Comment · Reviewer_2PvE · 2025-11-26
> > **Answer to rebuttal**
> >
> > I thank the authors for their rebuttal. However, I do not fully agree with the provided answers. Furthermore, some of my questions were not addressed.
> >
> > - **Related work:** The discussion in lines 98-103 is **insufficient**  and **ignores the majority** of the work in the field of conformal prediction for ITEs, potential outcomes, and counterfactuals, e.g., [1]-[6]. Without a proper discussion of this field, assessing the contribution and significance of the presented work is not possible in a fair way.
> >
> > -	**Various treatment types:** I do not agree with the authors. The provided derivations **do not hold** for, e.g., continuous treatments.
> >
> > -	**Novelty:** The authors stated that the paper by Candés 2023 was never used for their work. This is not true. The ideas of providing an LPB as well as the complete Theorem 4.2 are directly taken from this work (although obviously transferred to a slightly different setting).
> >
> > -	**Propensity estimation:** I am aware that many existing methods assume the propensity score to be known, which is a major weakness and only holds for RCT data. Here, the propensity score translates to the estimation of the function $w(x)$. Other works are similar to the reasoning in this paper and follow Lei & Candés (2021) by subtracting a term related to the *unknown* estimation error from the coverage guarantee, potentially rendering the final interval unreliable. Overall, the paper does not overcome any of the weaknesses in the former work regarding this issue.
> >
> > -	**Double robustness property:** This property is very similar to Thm. 2 in Candés 2023. Although I am aware that these properties have been stated and used before, I would like to highlight that **double robustness** is an asymptotic property, i.e., the **coverage is achieved asymptotically**, whereas **conformal prediction** is normally targeted at **finite sample guarantees** as is also claimed in this paper.  Therefore, the proposed method **does not provide valid intervals** if the weight function is misspecified, as confusingly claimed in the paper shortly before the Theorem. 4.2.
> > -	**Experiments:** I do not agree with the authors. The **proposed method does not outperform** the fused method in Figure 1. The new experiments in Figure 3 are neither showing a significant difference from the fused method. Overall, the provided **experiments are insufficient to evaluate the method** as already stated in my original review.
> >
> > Overall, I am still of the opinion that the proposed method is a mere **combination of two existing works** without a big own contribution besides translating it to a slightly different survival setting. The **empirical evaluation** does **not** underline the superior performance of the method. Finally, the formatting and citation errors are still not fixed. I am still of the opinion that the paper is **not yet ready for publication** and will keep my score.
> >
> > [1] Ahmed Alaa, Zaid Ahmad, and Mark van der Laan. Conformal meta-learners for predictive inference of individual treatment effects. In Conference on Neural Information Processing Systems (NeurIPS), 2023.
> >
> > [2] Zonghao Chen, Ruocheng Guo, Jean-Franc¸ois Ton, and Yang Liu. Conformal counterfactual inference under hidden confounding. In Conference on Knowledge Discovery and Data Mining (KDD), 2024.
> >
> > [3] Ying Jin, Zhimei Ren, and Emmanuel J. Candes. Sensitivity analysis of individual treatment effects: A robust conformal inference approach. Proceedings of the National Academy of Sciences of the United States of America, 120(6), 2023.
> >
> > [4] Jef Jonkers, Jarne Verhaeghe, Glenn van Wallendael, Luc Duchateau, and Sofie van Hoecke. Conformal Monte Carlo meta-learners for predictive inference of individual treatment effects. arXiv preprint, arXiv:2402.04906, 2024
> >
> > [5] Maresa Schröder, Dennis Frauen, Jonas Schweisthal, Konstantin Hess, Valentyn Melnychuk, and Stefan Feuerriegel. Conformal prediction for causal effects of continuous treatments. In Conference on Neural Information Processing Systems (NeurIPS), 2025.
> >
> > [6] Mingzhang Yin, Claudia Shi, Yixin Wang, and David M. Blei. Conformal sensitivity analysis for individual treatment effects. Journal of the American Statistical Association, 2022.

---

> > > ### Author Response · Authors · 2025-11-27
> > >
> > > Thank you for your further response. Below is our point-by-point reply to your comments.
> > >
> > > **Comment: Related works.**
> > >
> > > **Response:** The primary challenge of this work lies in the unknown censored data in the general right-censored setting for achieving marginal coverage. While there is a substantial body of work related to conformal prediction for ITE, and we sincerely appreciate the references you have provided, many of them did not consider the scenario of survival analysis. Our **Related Work** section will only discuss the advancements most directly relevant to the specific problem we address.
> > >
> > > **Comment: The terms "Multiple-treatment" and "Various treatment types".**
> > >
> > > **Response:** The term "multiple-treatment" in our work refers to the discrete setting. This is implied by the overlap condition, which means the positive probability of receiving each possible treatment. This condition cannot be satisfied in the continuous scenario.
> > >
> > > The additional experiments shown in Figure 2 demonstrate our method's ability to handle multiple treatments.
> > >
> > > **Comment: Novelty of our work.**
> > >
> > > **Response:** Theorem 4.2 is not directly applied from Candés (2023). Through the derivation of the inequality in Equation (1) of this paper, we utilize the non-conformity score for calibration, thereby eliminating the dependency on Type-I right-censored survival times. This enables us to propose a doubly robust method under general right-censored data.
> > >
> > > **Comment: Propensity estimation.**
> > >
> > > **Response:** The study of estimation error in the probability density function is orthogonal to the problem investigated in this paper. Besides, our work does not require the weight function to be known, and the doubly robust property of our method also compensates for the estimation error. Additionally, our empirical results showed that the estimation error is not a problem for us.
> > >
> > > **Comment: Double robustness property.**
> > >
> > > **Response:** To achieve finite-sample coverage, it typically requires that the calibration data and test data are exchangeable with each other. However, in a counterfactual setting, such exchangeable data is generally unavailable. Therefore, we must use weights to perform adjustments. The weight function itself needs to be estimated, making this an unavoidable issue—one that is also encountered in the existing literature. The primary focus of this paper is not on addressing the estimation accuracy under finite samples, but rather on how to implement the counterfactual conformal procedure in a general right-censored setting.
> > >
> > > **Comment: Performance illustrated from the experiments.**
> > >
> > > **Response:** We have conducted new experiments in Figure 3. The result shows that our method has the desired coverage in the presence of outlier data, while other methods with only PAC-type methods fail to do so.
> > >
> > > **Comment: Citation errors.**
> > >
> > > **Response:** Thank you for your suggestion. We have unified the reference formatting and corrected display errors caused by inconsistencies in the Bib file.

---

### Meta-Review · Area_Chair_bua7 · 2026-01-12

**Summary:**

This paper studies uncertainty quantification for counterfactual survival prediction under general right-censoring and proposes a conformalized framework for constructing lower prediction bounds with marginal coverage guarantees. By combining counterfactual quantile regression with weighted conformal calibration under strong ignorability, the authors extend conformal survival analysis to a setting that is substantially more challenging than prior work, which largely focuses on Type-I censoring or PAC-type guarantees. The theoretical analysis is careful and technically sound, and the empirical results on synthetic and real clinical data support the validity and practical relevance of the proposed approach.

While one reviewer raised concerns regarding novelty and the role of estimated weights in finite samples, I view these as conservative interpretations rather than fundamental limitations. The extension to general right-censoring with marginal validity for counterfactual survival bounds constitutes a meaningful contribution, and remaining presentation issues can be addressed in the final version. I therefore recommend acceptance.

**Reviewer Concerns:**

Several reviewer concerns were addressed by the rebuttal. In particular, reviewer 2PvE’s comments regarding notation errors, overstatements of guarantees, and ambiguity about the treatment setting were directly addressed through clarifications, corrections, and explicit discussion of the assumptions and scope of the method. The authors also expanded the related work discussion to better position the paper with respect to prior conformal prediction work in survival and counterfactual settings, addressing concerns raised by multiple reviewers. In addition, new experimental results examining robustness to outliers helped respond to questions about empirical validation.

Some concerns remain partially outstanding. Reviewer 2PvE’s skepticism regarding the degree of novelty relative to prior conformal counterfactual and survival work was not fully resolved and reflects a difference in interpretation rather than a remaining technical error. Similarly, the finite-sample impact of estimating the weight function remains an inherent limitation of weighted conformal approaches and is acknowledged but not eliminated by the rebuttal. These remaining concerns are primarily about emphasis and framing, rather than correctness or soundness of the proposed method.

**Reviewer Scores:**

Reviewer 2PvE.
Reviewer 2PvE expressed strong skepticism regarding novelty, finite-sample guarantees under estimated weights, and empirical superiority over baselines. While the rebuttal clarified several factual issues and added robustness experiments, the reviewer remained unconvinced. I expect Reviewer 2PvE would have maintained their score of 2, or at most increased it to 4.

Reviewer A1Xq.
This reviewer focused primarily on clarity, notation, and interpretation of the theoretical results. These concerns were addressed in the rebuttal. I expect this reviewer would have increased their score by one level, for example from 4 to 6.

Reviewer K7Lm.
Reviewer K7Lm viewed the problem as relevant and the methodology as sound, with moderate concerns about empirical validation. The rebuttal and additional experiments addressed most of these points. I expect this reviewer would have increased their score by one level, for example from 6 to 8.

Reviewer R9Tn.
This reviewer was generally positive about the technical contribution and theory, with minor presentation-related concerns. I expect this reviewer would have maintained a high score or increased it slightly, for example from 8 to 10.

Reviewer Q3Zd.
Reviewer Q3Zd was supportive of the overall contribution, emphasizing relevance and soundness, with only minor clarifications requested. I expect this reviewer would have maintained their score.

Overall, full discussion would likely have resulted in modest upward movement for most reviewers, with Reviewer 2PvE remaining the primary dissenting opinion.

---

### Decision · Program_Chairs · 2026-01-26

Accept (Poster)